# Verification of the System for Ship Position Keeping Equipped with a Set of Anchors in Unity3d

**DOI:** 10.3390/s22197421

**Published:** 2022-09-29

**Authors:** Jakub Wnorowski, Andrzej Łebkowski

**Affiliations:** Department of Ship Automation, Gdynia Maritime University, Poland Morska St. 83, 81-225 Gdynia, Poland

**Keywords:** dynamic positioning systems, anchor base positioning systems, Unity3d, game engines, marine systems, ship anchor winch modelling

## Abstract

Modern computers with specialised software are able to simulate oceans with waves and sea currents, and the action of wind, gravity, ships and other vehicles. The high-level programming languages that are used in this type of software can read information from navigation devices connected to the computer (e.g., via serial ports), and proceed to use the raw data in control algorithms. More and more desktop software and simulators can use data from additional electronic devices such as pressure sensors, temperature sensors, etc. Thus, it is possible to conduct real-time communication with a PLC (programmable logic controller) and use it in simulators. In this article, a user interface designed in Unity3d is presented. The user interface was able to read data from navigation devices, which were used in a ship positioning control algorithm. Verification of the algorithm occurred during research on a real ship, which used an anchor-based positioning system. Using data obtained on the real ship, a mathematical model of anchor winches was developed. Next, the mathematical model was implemented in the simulator developed in Unity3d. The simulator contained the same environmental conditions as during the research on the real ship. The mathematical model of anchor winches and implementation developed in the simulator will allow for future research on anchor-based positioning systems (e.g., in different environmental conditions). The research resulted in a shift of the ship’s position by 26.3 m under 280 degrees. The difference in arrival time to the target point between the real ship and the virtual ship was 19%, and the difference in position deviation was 330%.

## 1. Introduction

The most common type of ship positioning system is the dynamic positioning (DP) system, which uses tunnel thrusters, azimuth thrusters, cycloidal thrusters, and also propellers with or without a rudder, to keep position. The knowledge presented in publications about DP systems is very wide, ranging from methods for optimising ship motion [1] and ship controllers [2], to ship autonomy [3]. In DP systems, the target position can be determined by absolute or relative systems. In the first case, satellite navigation, i.e., GNSS systems, can be considered as an absolute system. This system is absolute because it directly determines the geographical position of the vessel. Positioning relative to a GNSS system is usually used for open water positioning. The second type of systems are called relative, because instead of a direct geographical position, distance and bearing to a target position are used for positioning.

Offshore vessels most commonly use microwave radar systems (MRS) as reference systems. There are several types of MRS, but each uses the same operating principle—a radar is placed on the ship to monitor transponders located on stationary objects (e.g., a platform). In addition to commercial MRS devices, solutions can be found in the scientific literature that use other dependencies related to microwaves. For example, the issue of increasing distance measurement accuracy using microwaves was discussed in [4]. The authors used linear frequency modulated (LFM) signals to obtain a position and radial velocity accuracy of 5.9 (cm) and 2.8 (cm/s), respectively, for moving objects. A general overview of the mathematical models used in microwave systems and the algorithms used therein to determine distance and bearing to a point, were presented in [5].

Another reference system is acoustic. Acoustic systems are mainly used for positioning ROVs (remotely operated vehicles) and AUVs (autonomous underwater vehicles). It consists of placing transponders on the seabed, to which the underwater vehicles position themselves relatively, by use of acoustic pulsation. Acoustic systems use several main techniques such as LBL (long baseline), SBL (short baseline) and USBL (ultra-short baseline). An acoustic system can be also used for ship positioning. To achieve this, a transponder is placed on the seabed, e.g., on pipelines or at other locations where positioning operations are carried out. The acoustic signals from transponders are received by an acoustic receiver located at the bottom of the ship. Issues related to the use of the acoustic LBL technique for position determination were presented in [6], and the issue of positioning using acoustic systems and triangulation techniques was presented in [7]. According to the results presented by the authors, a position accuracy of 10 (cm) was achieved for static objects.

Other reference systems that are commercially used for ship positioning are, additionally, laser distance measurement to a reference point (the problem of determining distance measurement limits for laser systems were discussed in [8], where it was shown that for underwater vessels, laser distance measurement works well at distances of 40–150 (cm); and in determining the distance of a vessel from a set point, the issue of laser distance measurement at high heeling was presented in [9]); the Taut-wire system, which is used less frequently due to operational problems; and inertial system (INS), which uses accelerometers to determine position [10].

This paper proposes a different way of positioning the vessel, i.e., using a set of anchors (the best position-keeping accuracy is obtained with four anchors, but three can be used with less position-keeping accuracy). The starting point in an anchor-based positioning system is to distribute the anchors evenly around the vessel at a distance of between 0.5 km and 1.5 km, depending on the available anchor rope length. Due to this distribution of anchor ropes, the system will not be applicable for positioning a ship close to platforms or other vessels, but is ideal for positioning over wrecks, and, due to the lack of working thrusters in the positioning process, is safe for divers. As no thrusters work during the positioning process, the system, unlike DP systems, does not consume energy during positioning. The position is kept by using anchor ropes only.

The concept of positioning a vessel using a set of anchors is not new. There are many ships in the world that use this system, but the whole process of placing anchors, positioning and changing the position of the ship, is performed manually by the crew. Although the system presented in this paper is not new, the knowledge presented in the scientific literature is poorer in comparison with the literature about DP systems. Publications focus on individual parts of the anchor-set positioning system, instead of presenting the system holistically. Problems related to the safety of anchored vessels have been presented in [11,12,13], and the problem of determining the depth of penetration of the seabed by a dropped anchor and the developed mathematical model of the anchor was presented in [14]. The problem of developing a mathematical model of a ship dropping anchor was discussed in [15]—unfortunately, the author simplified his model by depicting the dropping of only one anchor. As yet, there are no publications related to automation and control theory for this type of vessel.

The positioning algorithm proposed in this paper will allow the entire positioning process to be automated, and as a result, will help to improve safety on the ship and for people working under the ship.

As positioning using a set of anchors is very robust to changes in environmental conditions and maintains a very stable position, the algorithm presented later in this article focuses on changing the position of the ship once a new target position is given.

The operation of the proposed positioning algorithm is based on the determination of the lengths of the individual anchor ropes that need to be pulled in or loosened to bring the vessel to a new target position. For this purpose, the distance and direction to the target position must be determined and the geographical position of the individual anchors must be known. The positioning system proposed in [16,17] works in a similar way. The difference is, in those publications, the role of the anchors was carried out by sensors equipped with GPS modules, which reported their position in real time, while in this article, the anchors have a fixed geographical position assigned to them during the drop. Section 2 describes the proposed algorithm in detail.

## 2. Research Methodology

Ship positioning using anchors is one way of counteracting the external forces acting on the ship’s hull and keeping it in a preset position. When using thrusters for the same task, it is necessary to know the mathematical models of both the ship and the thrusters used. Such a system can be compared with a vertical plotter (Figure 1).

When considering the positioning of the vessel in the system shown in Figure 1, it is assumed that pulling in 1 m of rope moves the vessel by 1 m·*δ_L_*, where *δ_L_* is a coefficient depending on the stretch of the anchor rope. After changing the ship’s target position, new anchor rope lengths should be determined, to further move the vessel. The determination of the lengths of individual anchor rope can be obtained in several steps:Determine the distance and bearing to the target position, then calculate components of the move vector:
(1)x=d·sin(α)
(2)y=d·cos(α) 
2.Determine the distance and bearing between individual anchors and their anchor winches, then change these values into “*x*” and “*y*” components:
(3)$$Anchor1_{x}=d_{1}\cdot \sin(\alpha_{1})\;\;\;\;\;Anchor1_{y}=d_{1}\cdot \cos(\alpha_{1})$$
(4)$$Anchor2_{x}=d_{2}\cdot \sin(\alpha_{2})\;\;\;\;\;Anchor2_{y}=d_{2}\cdot \cos(\alpha_{2})$$
(5)$$Anchor3_{x}=d_{3}\cdot \sin\left(\alpha_{3}\right)\;\;\;\;\;Anchor3_{y}=d_{3}\cdot \cos\left(\alpha_{3}\right)$$
(6)$$Anchor4_{x}=d_{4}\cdot \sin\left(\alpha_{4}\right)\;\;\;\;\;Anchor4_{y}=d_{4}\cdot \cos\left(\alpha_{4}\right)$$
3.Determine of the distance between the anchor and the anchor winch in case the vessel has already moved to a new target position:
(7)$$d_{1}{}_{new}=\sqrt{{\left(Anchor1_{x}+x\right)}^{2}+{\left(Anchor1_{y}+y\right)}^{2}}\;\;$$
(8)$$d_{2}{}_{new}=\sqrt{{\left(Anchor2_{x}+x\right)}^{2}+{\left(Anchor2_{y}+y\right)}^{2}}\;\;$$
(9)$$d_{3}{}_{new}=\sqrt{{\left(Anchor3_{x}+x\right)}^{2}+{\left(Anchor3_{y}+y\right)}^{2}}\;$$
(10)$$d_{4}{}_{new}=\sqrt{{\left(Anchor4_{x}+x\right)}^{2}+{\left(Anchor4_{y}+y\right)}^{2}}\;$$
4.Determine new anchor rope length values:
(11)$$Anchor1_{newLine}=Anchor1_{actualLine}+\left(\delta_{L}\cdot \left(d_{1}-d_{1}{}_{new}\right)\right)\;$$
(12)$$Anchor2_{newLine}=Anchor2_{actualLine}+\left(\delta_{L}\cdot \left(d_{2}-d_{2}{}_{new}\right)\right)\;$$
(13)$$Anchor3_{newLine}=Anchor3_{actualLine}+\left(\delta_{L}\cdot \left(d_{3}-d_{3}{}_{new}\right)\right)$$
(14)$$Anchor4_{newLine}=Anchor4_{actualLine}+\left(\delta_{L}\cdot \left(d_{4}-d_{4}{}_{new}\right)\right)$$
where: *x, y*—vessel move values for “*x*” and “*y*” axis (m); *d*—distance to target point (m); *α*—bearing to the target point (°); *Anchor*1*_x_*, *Anchor*2*_x_*, *Anchor*3*_x_*, *Anchor*4*_x_*—distance between anchor and anchor winch for “*x*” axis (m); *Anchor*1*_y_*, *Anchor*2*_y_*, *Anchor*3*_y_*, *Anchor*4*_y_*—distance between anchor and anchor winch for ‘y’ axis (m); *d*_1…4_—distance between anchor and anchor winch (m); *α*_1…4_—angle between anchor and anchor winch (°); *d*_1*new*_…*d*_4*new*_—distance between anchor and anchor winch after ship moves (m); $Anchor1_{actualLine}$*,*
$Anchor2_{actualLine}$*,*
$Anchor3_{actualLine}$*,*
$Anchor4_{actualLine}$—actual rope lengths (m); $Anchor1_{newLine}$*,*
$Anchor2_{newLine}$*,*
$Anchor3_{newLine}$*,*
$Anchor4_{newLine}$—new rope lengths (m).

For the purpose of this research, simulation in Unity3d was used [19]. Unity3d has implemented components that allow the modelling of any kind of object and physical impact on these objects. The implementation of physics in Unity3d is based on conventional units, i.e., the physical values obtained depend on the values used for the calculations. For example, if the mass of an object is given in kilograms, then the force acting on this object must be in Newtons to obtain the object’s speed in metres per second. The simulator uses the units given in Table 1.

To carry out the research, emulators of navigation devices such as GPS and echosounder were implemented.

### 2.1. GPS

The GPS in the simulator worked by determining the distance and bearing to the vessel from the point (0,0). Then, using the WGS84 standard, the distance and bearing between the point (0,0) and the vessel, the geographical positions of the individual anchors were determined.

### 2.2. Echosounder

In Unity3d, the echosounder worked in a similar way to the real device. A beam termed “raycast” was sent from the location of the virtual echosounder; on collision with any object, the distance to this object was returned. Hence, information about water depth could be updated with the seabed level.

Devices such as a gyrocompass, MRU or log were not implemented because the information provided from them was available as the object’s properties.

In the simulator, the environmental forces were modelled on the basis of models provided by the classification society DNV GL (Det Norske Veritas Germanischer Llyod) [20]:Sea current force and torque;Sea waves force and torque.

### 2.3. Wind Force

The wind is the most important element of environmental forces, as it generates the biggest force on the ship’s hull, which results in a ship’s torque. The force from the wind has a two-dimensional value, as it acts on both the ship’s *X*-axis and *Y*-axis:(15)$$X_{wind}=q_{w}\cdot A_{FW}\cdot (-0.7\cdot \cos\left(\alpha_{wind}\right))\;$$
(16)$$Y_{wind}=q_{w}\cdot A_{LW}\cdot \left(0.9\cdot \sin\left(\alpha_{wind}\right)\right)\;$$

The torque generated by the wind was modelled using:(17)$$M_{wind}=Y_{wind}\cdot (x_{L}{}_{,air}+0.3\cdot \left(1-2\cdot \frac{\alpha }{\pi }\right)\cdot L_{pp}\;$$
where: *α_wind_*—wind angle relative to *L_pp_*/2 point (°); *A_FW_*—frontal projected wind area (m^2^); *A_LW_*—longitudinal projected wind area (m^2^); *x_L,air_*—longitudinal position of the area center of *A_LW_* (m); *L_pp_*—length between perpendiculars (m); *q_w_*—wind pressure on the ship’s hull factor.

The factor determining the wind pressure on the ship’s hull was described by the following function:(18)$$q_{w}=\frac{1}{2}\cdot \rho_{a}\cdot V_{W}^{2}\;$$
where: ρ*_a_*—air density (kg/m^3^); *V_w_*—wind speed (m/s).

### 2.4. Sea Current

The sea current does not generate as much force as the wind, especially in areas where its speed is marginal. The generated force was modelled as follows:(19)$$X_{current}=q_{c}\cdot B\cdot draft\cdot \left(-0.07\cdot \cos\left(\alpha_{current}\right)\right)\;$$
(20)$$Y_{current}=q_{c}\cdot A_{LC}\cdot \left(0.6\cdot \sin\left(\alpha_{current}\right)\right)$$

The torque generated by the sea current, was modelled using the following formula:(21)$$M_{current}=Y_{current}\cdot (x_{L,current}+\max\left(\min\left(0.4\cdot \left(1-2\cdot \frac{\alpha }{\pi }\right),0.25\right),-0.2\right)\cdot L_{pp}$$
where: *α_current_*—sea current angle relative to *L_pp_*/2 point (°); *A_LC_*—longitudinal projected submerged current area (m^2^); *x_L,current_*—longitudinal position of the area center of *A_LC_* (m); *B*—maximum breadth at water line (m); *draft*—summer load line draft (m); *q_c_*—sea current pressure on the ship’s hull factor; *L_pp_*—length between perpendiculars (m).

The coefficient for determining the pressure on the ship’s hull is analogous to the coefficient for wind force. It is required to change the air density value into water density value, and the wind speed into the sea current speed.

### 2.5. Waves Force

The sea waves force model depends on the used waves spectrum. The spectrum developed by Pierson Moskowitz was used for [21]. Based on this spectrum, the sea waves in our study were modelled in the following way:(22)$$F_{X}{}_{wave}=q_{waves}\cdot B\cdot h\left(\alpha_{waves},bow_{angle},\;C_{WLaft}\right)\cdot f\left(T_{surge}^{\prime }\right)\;$$
(23)$$h\left(\alpha_{waves},bow_{angle},\;C_{WLaft}\right)=0.09\cdot h_{1}\left(\alpha_{waves},bow_{angle},\;C_{WLaft}\right)\cdot h_{2}\left(\alpha_{waves}\right)$$
(24)$$h_{1A}\left(bow_{angle}\right)=0.8\cdot bow_{angle}^{.45}$$
(25)$$h_{1B}\left(C_{WLaft}\right)=0.7\cdot C_{WLaft}^{2},\;C_{WLaft}\in \left[0.85,1.15\right]$$
(26)$$dir\left(\alpha_{waves}\right)=\{\begin{array}{c} \alpha_{waves},\;0\leq \alpha_{waves}\leq \pi \\ 2\pi -\alpha_{waves},\;\pi \leq \alpha_{waves}\leq 2\pi \end{array}$$
(27)$$h_{1}\left(\alpha_{waves},bow_{angle},C_{WLaft}\right)=h_{1A}\left(bow_{angle}\right)+\frac{dir\left(\alpha \right)}{\pi }\left(h_{1B}\left(C_{WLaft}\right)-h_{1A}\left(bow_{angle}\right)\right)$$
(28)$$h_{2}\left(\alpha_{waves}\right)=0.05+0.95\cdot \tan^{-1}\left(1.45\cdot \left(dir\left(\alpha_{waves}\right)-1.75\right)\right)$$
(29)$$f\left(T^{\prime }\right)=\{\begin{array}{c} 1,\;if\;T^{\prime }<1\\ {T^{\prime }}^{-3}\cdot e^{1-T^{\prime^{-3}}},\;if\;T^{\prime }\geq 1 \end{array}$$
(30)$$F_{Y}{}_{wave}=q_{waves}\cdot L_{OS}\cdot \left(0.09\cdot \sin\left(\alpha_{waves}\right)\right)\cdot f\left(T_{sway}^{\prime }\right)$$
(31)$$M_{z,wave}=F_{Y}{}_{wave}\cdot (X_{Los}+\left(0.05-0.14\cdot \frac{dir\left(\alpha_{waves}\right)}{\pi }\right)\cdot L_{OS}$$
(32)$$T_{surge}^{\prime }=\frac{T_{z}}{0.9\cdot L_{pp}^{0.33}}$$
(33)$$T_{sway}^{\prime }=\frac{T_{z}}{0.75\cdot B^{0.5}}$$
(34)$$H_{s}\left(V_{w}\right)=0.3125\cdot V_{w}-0.62$$
(35)$$T_{z}\left(V_{w}\right)=0.741\cdot V_{w}+0.536$$
(36)$$q_{waves}=\frac{1}{2}\cdot \rho_{w}\cdot g\cdot H_{s}^{2}$$
where: *H_s_*—significant wave height (m); *L_os_*—longitudinal distance between the fore-most and aft-most point under water (m); *L_pp_*—length between perpendiculars (m); *X_Los_*—longitudinal position of *L_os_*/2 (m); *bow_angle_*—angle between the vessel *x*-axis and a line drawn from the fore-most point in the water line to the point at *y* = *B*/4 on the water line (°); *C_WLaft_*—water plane area coefficient of the water plane area behind midship (-); *α_waves_*—waves coming from direction (°); *B*—maximum breadth at water line (m); *g*—standard gravity (m/s^2^); ρ_w_—water density (kg/m^3^); *V_w_*—wind speed (m/s); q_waves_—wave pressure on the ship’s hull factor.

In order to clarify some of the designations in Formulas (15)–(36), the following Figure 2 may be helpful.

### 2.6. The Ship Model

In the developed simulator, the ship was modelled as a solid object submerged in water, whose displacement was represented by the equation:(37)$$D=\rho_{w}\cdot V_{s}{}_{w}\cdot g$$

The weight of the ship was modelled as:(38)Q=m·g

The following condition must be passed for the ship to stay afloat:(39)D=Q

In addition to the resistance associated with the effects of wind force and sea current on the ship’s hull, the friction resistance of the hull was also modelled:(40)$$R_{v}=\frac{1}{2}\cdot \rho_{w}\cdot C_{x}\cdot C_{F}\cdot A_{LC}\cdot V^{2}$$
where: *R_v_*—friction resistance; *D*—ship displacement (kg); *ρ_w_*—water density (kg/m^3^); *V_sw_*—volume of the submerged part of the hull; *g*—standard gravity (m/s^2^); *Q*—ship weight (kg); *m*—ship mass (kg); *C_x_*—hull shape coefficient (-); *C_F_*—friction coefficient for steel (-); *A_LC_*—submerged hull surface (m^2^); *V*—ship speed (m/s).

The outcomes of the effects of environmental forces on the ship’s position and the hull’s frictional resistance during movement are described in Section 3.4: The Environmental Forces and the Friction Resistance of Water.

## 3. Verification of the Mathematical Model of Anchor Winches in Unity3D

To verify the ship’s position-keeping system in Unity3d, a model of the ship, ropes and known weather conditions was needed for simulations in the virtual environment. The research described in this article was carried out on a vessel with the dimensions shown in Table 2.

Hydraulic pumps were attached to each pair of anchor windlasses (bow pair and stern pair) to generate pressure in the hydraulic system. Due to the low power consumption during positioning, only one pump worked on the bow, and one on the stern. Together, the pumps drew 348 kW from the generator. Figure 3 shows a simplified diagram of the connection between the generator and the hydraulic pumps.

The vessel was equipped with four anchors and four hydraulic anchor winches—two at the bow and two at the stern. Each was equipped with 1500 m of steel cable. With this anchor arrangement, the vessel could resist environmental forces from any direction, provided they were all in use. The details of the anchors’ winches’ position on the ship are shown in Table 3 and Figure 4.

The anchor winches were controlled via a PLC. Data from the individual anchor winches were displayed on the operator’s computer via the user interface (Figure 5).

The user interface, shown in Figure 5, was developed in Unity3d (using C# language [23]) to display the basic data received from the navigation devices, the data of the individual anchor winches, and to visualise the movement of the vessel.

The navigation devices were connected to the operator station via RS232 serial ports with the following parameters: speed—4800; data—8 bit; parity—none; stop bit—1 bit.

The data read from each device were implemented as separate threads in order to eliminate delays and freezes in the user interface. Data were loaded into the system on arrival to the serial port input in a format suitable for the NMEA 0183 protocol. The refresh rate of the displayed data in the interface was every 500 ms. The exception was the ship’s course, which, due to the ship’s shape animation, needed to be refreshed as often as possible; in this case, a refresh rate of 130 times per second was achieved (this result was redundant, but did not affect system performance). The second exception was the depth chart. Adding a new depth value to the chart occurred once per second. In addition, the operator had the option of determining how long the depth change history should be displayed. The user could select 1, 5, 15, 30, or 60 min.

Communication with the PLCs of the individual anchor winches was via the Modbus TCP/IP protocol and using the EasyModbus library. As with the navigation equipment, individual threads were used to exchange information with the controllers. Using individual threads, an information exchange time of 200 ms was achieved without any loss in system performance. The longest time the system ran with the interface enabled, was one month. After this time, there was no increased CPU usage or increased RAM consumption.

An enlarged user interface is shown in Figure 6.

The user interface was divided into two sides. On the left side were the following items, starting from the top:System alarms—displayed the last four alarms. The next alarms were visible after expanding the list of alarms. The alarm could be accepted from the application or the system operator’s keypad;Ship speed—longitudinal and transverse speed;Actual position and target position;Tilts;Depth chart;Actual course, ROT;Information about individual anchor/anchor winch:○Distance between ship and dropped anchor;○Anchor’s depth;○Actual rope length;○Target rope length;○PLC connection indicator.


On the right side of the interface were:Shape of the vessel;Information about wind speed and direction;Information about the value and direction of the resultant force from the anchor ropes tension.

The research on the real ship took place in Gdansk Bay [24] on 19 May 2022. Figure 7 and Figure 8 show the hourly changes in wind speed and direction during the research period.

The average value of the sea current speed in Gdansk Bay on the same day, was 15.4–35.2 (cm/s), that is, approximately 0.15–0.35 (m/s) (Figure 9).

### 3.1. The Research on the Real Ship 

The research on the real ship consisted of verifying the algorithm described by Equations (1)–(14), in accordance with the following:That four anchors be used in the research;That the deviation of the position after changing the target point should be less than 5 m.

For this purpose, a new anchorage plan was designed before starting positioning operation. To create the anchorage plan, a tool developed in the Unity3d was used (Figure 10). The tool is attached to the user interface from Figure 6.

The anchorage planning tool consisted of a geographic grid and a settings window which contained the anchorage parameters.

The information that could be found in the settings window was in the following order:Date of project creation;Name of project;Project description;Geographical coordinates of the target point;Expected depth at the target point;Environmental conditions:○Expected wind speed and direction;○Expected speed and direction of the sea current;○The proposed course of the vessel by the tool. The tool always proposed the same course as the wind direction;○Ship’s course selected by the operator.


It was important to declare which anchors were to take part in the positioning operation. After selecting the anchor, a green area was shown, indicating the range of safe anchor drop. The area was defined by the maximum rope length that was available on the anchor winch. When the anchor was dragged to a location within the green area, the geographical coordinates of the anchor were stored.

For security reasons, the exact geographical coordinates of the ship have been blurred.

Based on the anchorage plan presented above, a ship’s track was generated. The track was loaded into the system, which created the ship’s trajectory. The anchor drop order was represented by the number next to the anchor, e.g., **BP (1)** meant that the anchor should be dropped first.

Once the anchoring operation was completed and the ship’s position was stabilised, the set-up of the specific anchorage elements was as shown in Figure 11:

The distance of each anchor from the vessel and the initial ropes lengths are shown in Table 4.

The ship’s course was 312 degrees, while the averaged wind angle of attack on the hull was 100 degrees relative to north.

The carried out research was an attempt to move the vessel 23.6 m at an angle of 280 degrees to the north using only the anchor winches (Figure 12).

A diagram describing the system operation and flow of information in the positioning algorithm is shown in Figure 13.

The first step after setting the new target position was to recalculate the anchor rope lengths according to Formulas (1)–(14). By substituting the geographical position of the vessel, the geographical position of the individual anchors and the distance to the new target position, the values given in Table 5 were obtained.

The next step was obtaining the anchor rope lengths, given in Table 5. The control of the individual anchor winches was conducted in such a way that the anchor winches that pulled in the rope achieved this with a small step (0.1–0.5 m/s), while the anchor winches that loosened the rope had a fixed tension set. The fixed tension on these winches meant that if the actual tension of the anchor rope was less than the declared tension, the winch did not work. If the tension increased above the declared value, the winch loosened the rope until the tension fell below the declared value. With this control strategy, the vessel moved slowly (about 0.1 m/s, and the whole operation took more than 3 min), but it was possible to achieve a straight-line arrival at the set point without any oscillations around it (Figure 14).

The point for time equal to 0 corresponded to the moment when the recalculation of the rope lengths by the positioning algorithm took place. The movement of the vessel started after about 10 s after the new anchor rope lengths were determined, which was the time needed to pull in enough BP and BS anchor ropes to increase the tension on the other ropes. Figure 15 shows that at the same time, the other anchor winches did not make any movement. As anchor lifts SP and SS started to loosen the anchor ropes, the vessel started to move towards the target point. The target point was reached in approximately 158 s after the anchor rope lengths were recalculated. After this time, all anchor winches were stopped. Due to the effect of compensating tension, the vessel moved away from the target point for approximately 0.6 m and then returned closer again by a distance of 0.1 m.

Changes in the length of individual ropes during the operation are shown in Figure 15.

In the graph above, the largest participation in the vessel’s move was the **BP** anchor winch, which pulled in the anchor rope evenly. The anchor winch **SS**, which was on the opposite side of the vessel, had the task of introducing resistance to the vessel. Due to this resistance, the ship did not gain too much speed. The **SP** and **BS** anchor windlasses had lesser roles due to the small value of the difference between the initial value of the anchor rope length and the target rope length, for each windlass.

According to Figure 3, during operation, the hydraulic pumps drew 348 kW from the generator. Anchor positioning with a hydraulic system is a specific type of positioning because energy is only consumed for the hydraulic pumps, and only when the position of the vessel needs to be changed. When the ship is not making any movement, the pumps remain off and the energy consumption is zero. This is an advantage over the DP system, where the thrusters must operate all the time, whether the vessel is moving or not; it makes no difference whether the anchor winches are pulling in the rope, loosening the rope, or running faster or slower, the power consumption for the pumps is the same. Knowing the power input from the generator (348 kW) and the time taken to move the ship by 26.3 (m) (158 s ≈ 3 min), it was estimated that the ship consumed about 17.4 kWh of energy [26]. Another advantage of the anchor-based positioning system is that there was no need to use external reference systems (laser, ultrasonic, radio, radar, etc.) to position the ship.

### 3.2. The Simulation

The research on the real ship was undertaken to collect the characteristics of the anchor winches in different situations, in order to determine their mathematical model. The next step was to verify this model in the simulator developed in Unity3d using the following assumptions:The environmental conditions in Unity3d should be the same as during the research on the real ship;The virtual ship must have the same dimensions as the real ship;The geographical coordinates of the virtual ship and the dropped anchors must be the same as during the research on the real ship;The position of the virtual ship must be shifted in the same way as the real ship;The mathematical model of the anchor winches should represent the work of the anchor winches on the actual ship.

The graphical interface of the simulator is shown in Figure 16.

The user interface, as shown in Figure 16, consisted of the following parts:Indicators showing if the anchor had been dropped; if so, it displayed the current rope length and tension;The shape of the vessel;Indicator of actual course of the vessel;A window used to specify the target position, and a window with the actual distance and bearing to the target position.

### 3.3. The Ship Model

In Unity3d, an Advanced Ship Controller component was used to model the ship and its behaviour on the water. The main window of the component is shown in Figure 17.

The component used the geometry of the 3D model to determine parameters such as buoyancy, heeling or the effect of water friction on the hull, so an accurate 3D model of the ship was required. If a ship is 73 m long, then the model should also be 73 units long. If the dimensions of the ship do not match those of the 3D model, this could result in automatic sinking due to excessive weight.

In addition, the component also allowed the modelling of engines, tunnel thrusters and propellers with rudders. The windows with the models’ settings are shown in Figure 18.

In order for the “Advanced Ship Controller” component to be able to have an effect on the physics of the ship, an additional “Rigidbody” component was added to the object (Figure 19). This component turned on the physics affecting the object.

The “Rigidbody” component included following properties:Mass—mass of the object;Drag—determination of air resistance for the moving object (0—no air resistance);Angular drag—determination of the air resistance during rotation of the object.

### 3.4. The Environmental Forces and the Friction Resistance of Water

The environmental forces that affected the vessel were modelled using Equations (15)–(36). The direction and speed of the wind are shown in the graphs in Figure 20 and Figure 21.

The above charts were plotted from data read from the wind sensor on the real ship during the research. The wind direction shown in Figure 21 was the true wind, and due to this fact, the wind was converted to be relative to the ship.

The speed of the sea current was taken as an average value of 0.25 (m/s) based on the values shown in Figure 9. The direction of the sea current was taken as the wind direction.

Unity3d has several methods for interacting with a given force on an object. For example, there are methods from the “Rigidbody” component, such as AddForce(), AddRelativeForce(), AddTorque(), AddRelativeTorque(), but their disadvantage is that they interact with the object “impulsively”, i.e., one call of a method is equivalent to one push of the object with a given force. A second way to interact with an object with a given force is to use the “Constant Force” component (Figure 22).

The Constant Force component is characterised by the fact that it acts on an object as long as the value of the force does not change, therefore, it was used to simulate the effect of environmental forces on the virtual ship and to simulate the frictional resistance of the hull. The values of *X*, *Y*, *Z* were the force values acting on the individual axes of the object. If the mass of object in “Rigidbody” component was given in kilograms, then the force value was expressed in Newtons.

### 3.5. Tension

The “Fixed Joint” component was used to determine the tension of the rope (Figure 23).

The “Fixed Joint” component returned the value of the force with which the object was pulled. The working component is illustrated in Figure 24.

### 3.6. Anchor Ropes

The “Obi Rope” component was used to simulate anchor ropes. This component allowed the generation of ropes, chains, cables, etc., and allocated them the properties of the corresponding material. Figure 25 shows the properties that can be set for ropes.

The “Obi Rope” component created ropes consisting of small particles. Each property, shown in Figure 25, had its own setting fields:Evaluation—determined whether a property was added one at a time to each particle, or to all of them at the same time;Iterations—how many times per second the particles were updated;Relaxation—how much influence the property had on particles (1 = 100%, but 200% or more could be set).

Properties in the “Constraints” field were related to interactions between particles, e.g.,:Distance—determined how far the particles could move away from each other, so it determined the stretchability of the material;Bending—determined how much the particles could bend, so it determined the stiffness of the material;Pin—determined how far the rope could move away from the point of attachment under tension;Chain—checking this option meant that the rope was treated as a chain. The rope length did not change under tension.

The following properties were used in the simulation:Distance: evaluation—sequential; iteration—100; relaxation—1.5;Pin: evaluation—sequential; iteration—200; relaxation—1;Chain: evaluation—sequential; iteration—200; relaxation—2.

These properties were the best for the steel anchor rope. Due to the imperfection of the physics engine, it was not possible to obtain inextensible materials; there was a minimal stretching effect of the rope under tension every time. This limitation was not only present in Unity3d, but will be present in any game’s engine.

### 3.7. The Anchor Winch Control

Unity3d is equipped with several main functions, including the ***Update()*** function, which calls itself every frame. For example, if an application has 60 fps, the ***Update()*** function will call itself 60 times per second. This property was used to control the individual anchor winches, as all the calculations, tension control and rope length control were inside it.

The general control algorithm is shown in Figure 26.

The whole control process started with the new positioning point. The lengths of the anchor ropes were recalculated according to Formulae (1)–(14). After recalculation, the new lengths of anchor ropes were passed on to the individual anchor winches, which worked according to the algorithm shown in Figure 27. After the vessel moved, the position was verified. If the achieved position was not satisfactory, the lengths of the ropes were recalculated.

First, the control algorithm located on each anchor winch determined the difference between the declared anchor rope length value and the current value. If difference was smaller, it meant that the rope was pulled in by the declared step, e.g., by 0.1 m, and then the difference was checked again. If the difference was greater, it meant that the rope was loosened, but only if the tension value exceeded the declared value.

The first verification of the above control method was performed using a simple example of rope pulling. The entire system is shown in Figure 28. Two winches were placed opposite each other at a distance of 100 m. A ball was inserted between them. The ball was connected to the individual winches by two pieces of rope.

The experiment consisted of moving the ball 20 m to the right at a speed of 0.2 m/s. To simplify, the movement of the ball along the *X* and *Y* axes was blocked, so the ball always moved in a straight line. The changes in the length of the individual ropes are shown in the graph in Figure 29.

During the experiment, the left winch started to loosen the rope 3 s after the start of the algorithm, only when the rope tension had risen above the declared value. In the next seconds, as both winches were running at the same speed, the tension was kept constant. Therefore the left winch loosened the rope at same moment as the right winch pulled in the rope. When the right winch finished its work, due to the stretching effect and the tension value falling below the declared value, the left winch did not end up loosening 20 m of rope, it completed the work by loosening 19.6 m of the rope.

After positive verification of the anchor winch control system, the next research was carried out on the virtual ship model. During the scene set-up in Unity3d, an attempt was undertaken to replicate the ship and anchor positions, along with the same anchor rope lengths and initial tensions, from the research, on the real ship. The initial set-up is shown in Figure 30. The environmental forces that acted on the vessel during the research on the real ship also worked during simulation.

During the test, the virtual vessel was moved in the same way as the real vessel—23.6 m at an angle of 280°. After specifying new target position, the process of determining the new rope lengths was conducted according to the algorithms from Figure 26 and Figure 27. The changes in the distance of the vessel to the target point during the move are shown in the graph in Figure 31.

The movement of the vessel started 8 s after the anchor rope lengths were recalculated. The anchor winches that pulled in the rope achieved this with a constant step of 0.1 m/s. This caused the vessel to slowly reduce the distance to the target point until, after 4 min, the vessel stopped approximately 2 m from the target point, when the anchor winches stopped working. The changes in the individual anchor rope lengths during the move are shown in Figure 32.

The **BP** and **BS** anchor winches pulled in the anchor ropes at 0.1 (m/s) evenly over time. Anchor winches **SP** and **SS**, in response to the increasing tension, loosened the ropes only when the tension value was above declared value. In this way, the vessel did not gain too much speed. The disadvantage of this solution was that, due to too low speed, the ship was unable to reach the target point.

## 4. Comparison of Real Ship’s Characteristics with Characteristics Obtained in the Unity3d

Modelling the hydraulic anchor winches that were on the real ship was not possible in Unity3d due to the inaccuracy of the sensors used. The measurement of the rope length was obtained using an optical encoder, which sometimes lost steps. For this reason, the anchor winches operated once faster and once slower. The constant rope tension controller was used as a hydraulic bypass valve, which sometimes returned a different value, which caused the anchor rope to spontaneously be pulled in or loosened.

Due to the random nature of the above errors, in Unity3d, anchor winches were modelled to work slower than anchor winches on a real ship, but were free of errors. Figure 33 compares the ship’s position deviation after changing target position, between the real ship and the virtual ship.

In the case of the real ship, it took about 188 s to reach the target point, while for the virtual ship, it took about 233 s. The difference in move time was 45 s. This was due to the difference in the speed of the individual anchor winches. On the real ship, the anchor winches operated using hydraulics, and, therefore, the speed of pulling in or loosening the rope and the tension held was dependent on the pressures in the system. Any pressure fluctuation caused the anchor winch to operate once faster and once slower, especially at random moments. For this reason, in the simulation it was decided to use a constant speed for the anchor winches, which resulted in a slower time to reach the target point.

The slower operation of the anchor winches also resulted in differences in the changes of the individual rope lengths during the ship’s move. The comparisons of differences for specific ropes are shown in Figure 34, Figure 35, Figure 36 and Figure 37.

The **BP** rope for both the real ship and the virtual ship had the task of pulling in 20.5 m of rope. In the case of the model in Unity3d, the winch worked evenly with a step of 0.2 m/s, while on the real ship, the rope pulling-in speed varied between 0.1 m/s and 0.5 m/s, which resulted in a faster pull-in of rope.

The **BS** anchor winch, similar to the **BP**, had the task of pulling in a specific length of rope. Again, for the virtual model, the anchor winch operated at a constant speed of 0.2 m/s and finished in about 118 s, while the anchor winch from the real ship finished in about 161 s. Time difference was 43 s.

The task of the **SP** anchor winch was to loosen 5.8 m of anchor rope. In the virtual environment, the anchor winch loosened the rope at a speed of 0.2 m/s, which was similar to the loosening speed on the real ship. The anchor winch located on the real ship and on the virtual ship both finished their work after about 50 s, but the anchor winch located on virtual ship loosened more rope—about 20 cm.

The SS anchor winch had a task to loosen 24.5 m of rope. Again, in the virtual environment, a winch operating speed of 0.2 m/s was used, resulting in the entire operation being completed in approximately 200 s. On the real ship, the same operation was completed in about 160 s. The difference was 40 s.

The research on the real ship was aimed at developing a mathematical model of the anchor winches and then implementing it in the simulator. A mathematical model of anchor winches will allow additional research (for example, in different environmental conditions) to be performed without access to the real ship.

The mathematical model of anchor winches presented in this article did not represent the full performance of the positioning system on the real ship. The simulator did not implement the distortions caused by the imperfect sensors. Hence, the mathematical model was too ideal. Figure 35 and Figure 36 show the differences between characteristics obtained from the ideal mathematical model and from the real ship. Another reason for difference between the simulated model in Unity3d and the real ship was the too highly stretchable rope in Unity3d. Stretchable rope prevented the ship from reaching the target point because the rope could not build enough tension. Hence, the anchor winches turned off too fast. The accuracy of the measurement of the geographical coordinates was a further reason for difference. On the real ship, the GPS position accuracy was about 1 m, while the Unity3d position resolution was about 1 mm.

## 5. Summary

Currently, on ships using an anchor-based positioning system, changing the ship’s position is performed manually by the crew. The operator manipulates the lengths of the individual anchor ropes using a control panel for each anchor. The use of four anchors in the positioning process means that the operator must focus on operating four separate joysticks. In addition, a person must be at each anchor winch to keep an eye on the anchor rope, which can cause dangerous situations, such as rope tangling around a leg.

Automating the ship control process through the development of a positioning algorithm makes it possible to reduce the workload on the anchor winch operator and improve work safety by excluding the human element.

This article presented the concept of an automatic positioning system using a set of anchors, which consisted of the following components:Operator station with a user interface developed in Unity3d;Algorithm that determined the lengths of individual anchor ropes based on the actual geographic position of the ship, the geographic position of the anchors and the geographic position of the target point. The change of the determined anchor rope lengths caused the ship to move to a new target point;The control algorithm for individual anchor winches, located on PLCs.

Testing the algorithms’ accuracy was carried out on a real ship that used a set of anchors during positioning. The algorithm that determined anchor rope lengths was tasked with determining anchor rope lengths that would move the ship 26.3 m at an angle of 280 degrees. According to the characteristics shown in Figure 16, it took about 158 s to reach the target point for the first time.

In order to test the manoeuvring ability of a ship using an anchor system and to determine limit values for hydrometeorological conditions, a mathematical model of anchor winches was developed. The model was implemented in a simulator, which was developed in Unity3d. The next step was to conduct the same tests under virtual conditions, as on the real ship, to verify the accuracy of the model. According to the obtained characteristics, from Figure 31, of reaching the target point, the arrival time was 233 s. The difference between the arrival time of the real ship and the virtual ship was 19%. Differences in target point arrival times were caused by the following:Too perfect representation of the work of anchor winches in the mathematical model;The anchor ropes in the mathematical model had a stretch effect because non-stretchable ropes could not be obtained;In the simulator, there was a higher precision in the calculation of geographic coordinates. On the real ship, the system operated on the geographic position provided by the GPS with an accuracy of up to 1 m, while the simulator determined the geographic position with a resolution of 1 mm.

Evaluating the quality of control in the case of positioning using a set of anchors was difficult, since no standards have been developed to define how accurate the position should be kept by this type of system. On the other hand, if standards for dynamic positioning systems are taken as an assessment of control quality, classification societies such as DNV GL, when inspecting DP systems, allow for deviation from target position of up to 5 m for DP 2 systems. The algorithm presented in this article was implemented on the ship where the research was conducted. By using this algorithm, a reduction in the number of people required to operate the system possible. For example, a radar user, when noticing an object over which they want to position, would not need to notify the person responsible for operating the anchor winches, but could simply enter the new geographic coordinates of the target point into the system.

In addition to the advantages of the developed algorithm mentioned above, the presented system also allows for the automation of the anchoring process, thus, reducing the time needed to place anchors. In addition, the algorithm checks whether the set point has changed. When a change in the coordinates of this point is detected, the process of moving the vessel to the position of the new target point is automatically initiated. The current position of the vessel is also monitored. If the deviation from the target point position is greater than the value declared by the operator, the vessel’s position is automatically corrected. Another characteristic of the system is that reference systems are not required to keep the position, as is the case with DP systems.

Unfortunately, the algorithm developed to control the anchor-based positioning system has some disadvantages. The main one is that with the use of two anchors, the stabilisation of the ship’s position is ineffective. In addition, the use of four anchors are required to change the ship’s course. A definite disadvantage of the anchor-based positioning system in relation to DP systems is the slow process of placing anchors, changing the ship’s position and shifting from a designated position.

## Figures and Tables

**Figure 1 sensors-22-07421-f001:**
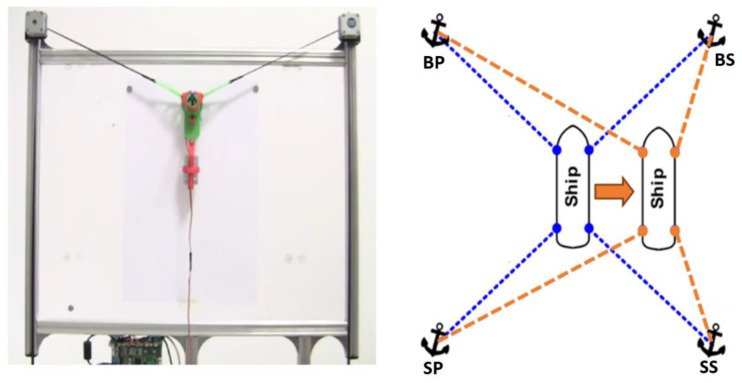
Comparison of a vertical plotter [18] with anchor-based positioning.

**Figure 2 sensors-22-07421-f002:**
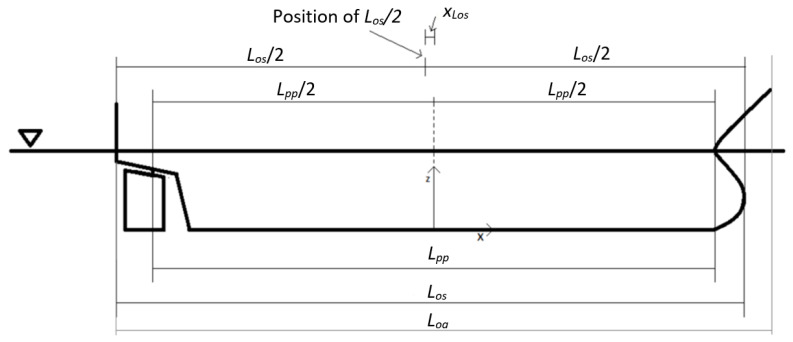
Ship geometrical parameters [22].

**Figure 3 sensors-22-07421-f003:**
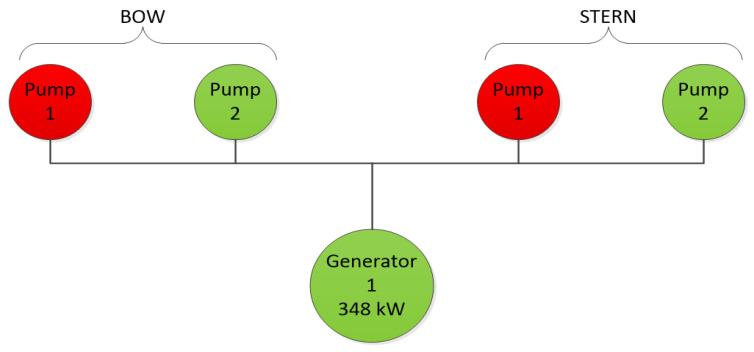
Generator–pump connection diagram. The green colour of the pump indicates that the pump is in use, and the red colour indicates an inactive pump.

**Figure 4 sensors-22-07421-f004:**
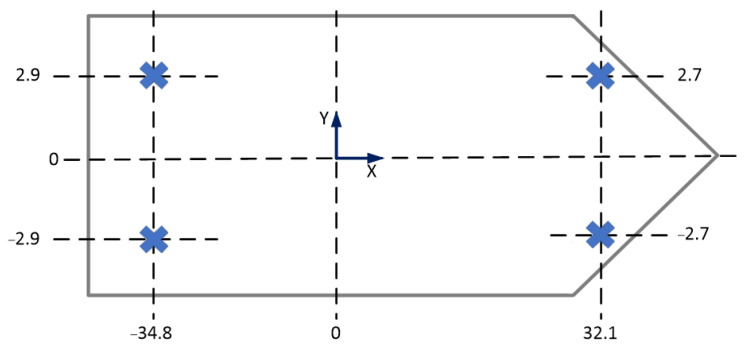
Detailed arrangement of anchor winches on the vessel.

**Figure 5 sensors-22-07421-f005:**
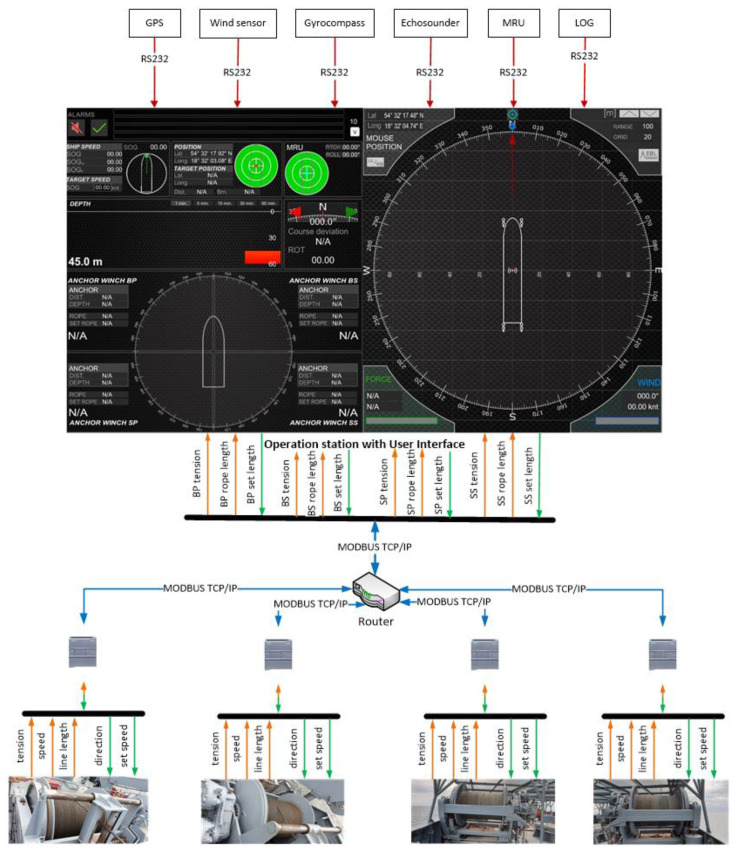
Diagram of the anchor-base positioning system.

**Figure 6 sensors-22-07421-f006:**
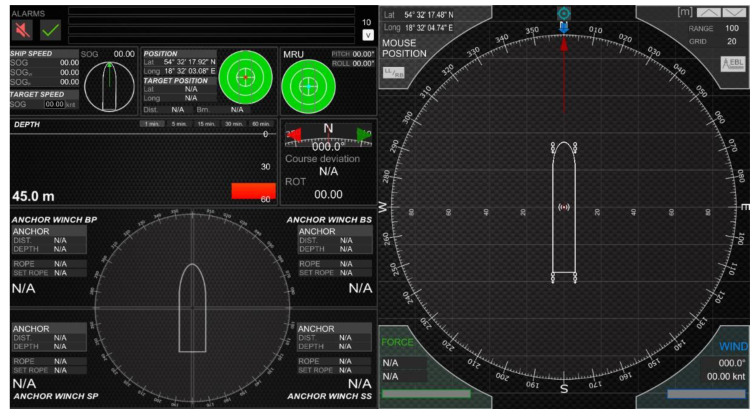
User interface developed in Unity3d.

**Figure 7 sensors-22-07421-f007:**
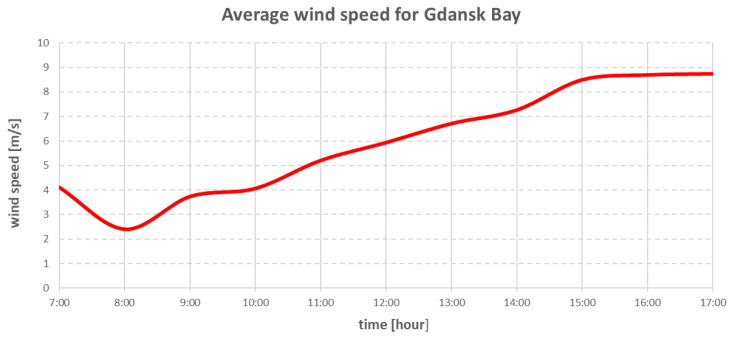
Averaged wind speed changes in Gdansk Bay on 19 May 2022.

**Figure 8 sensors-22-07421-f008:**
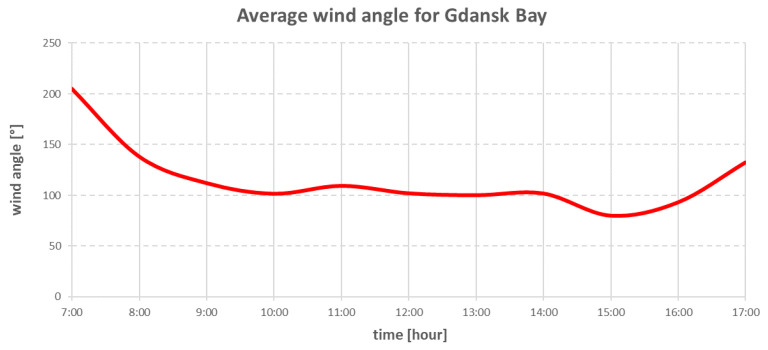
Averaged wind angle changes in Gdansk Bay on 19 May 2022.

**Figure 9 sensors-22-07421-f009:**
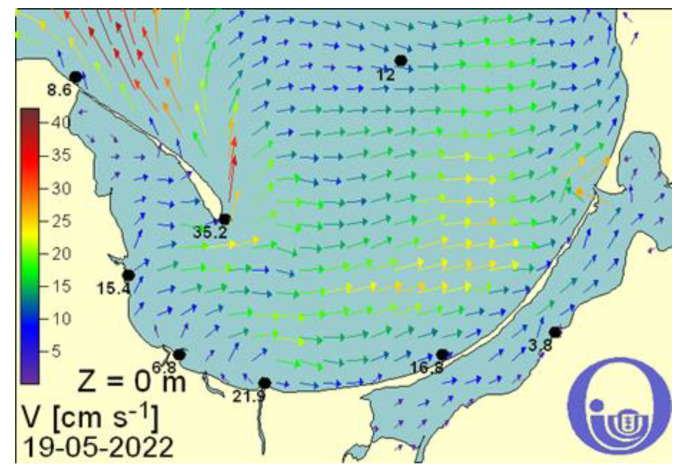
Sea current speed values for Gdansk Bay on 19 May 2022 [25].

**Figure 10 sensors-22-07421-f010:**
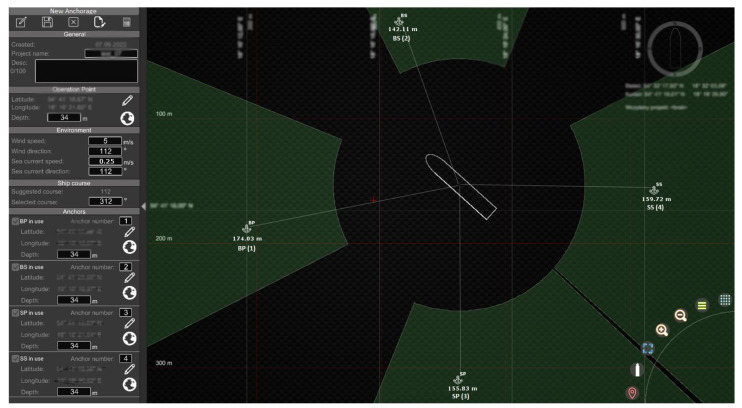
Anchorage planning tool developed in the Unity3d.

**Figure 11 sensors-22-07421-f011:**
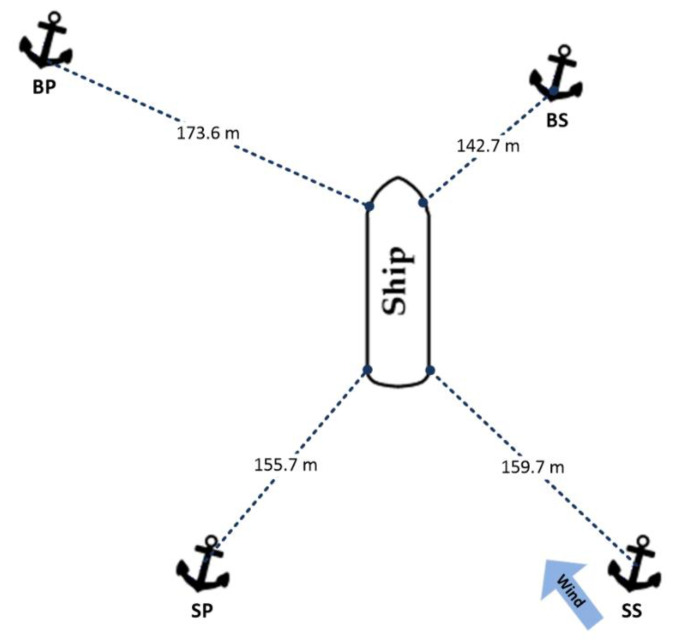
Anchor arrangement and vessel position during positioning.

**Figure 12 sensors-22-07421-f012:**
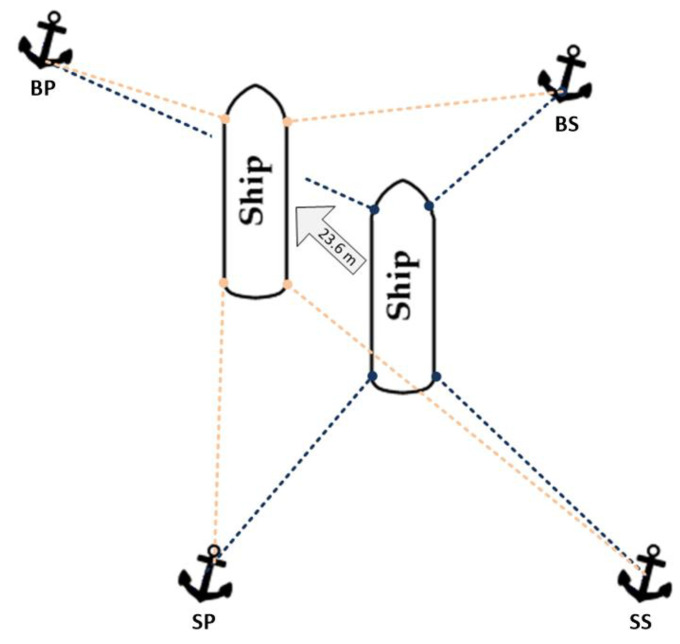
Visualisation of the ship’s trajectory.

**Figure 13 sensors-22-07421-f013:**
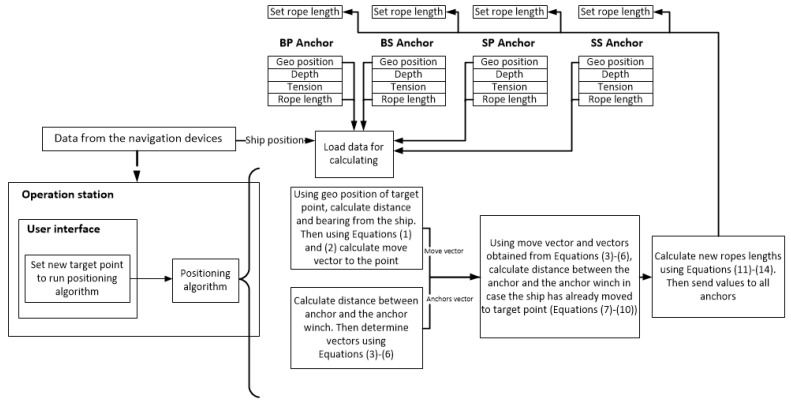
Information flow in the positioning system.

**Figure 14 sensors-22-07421-f014:**
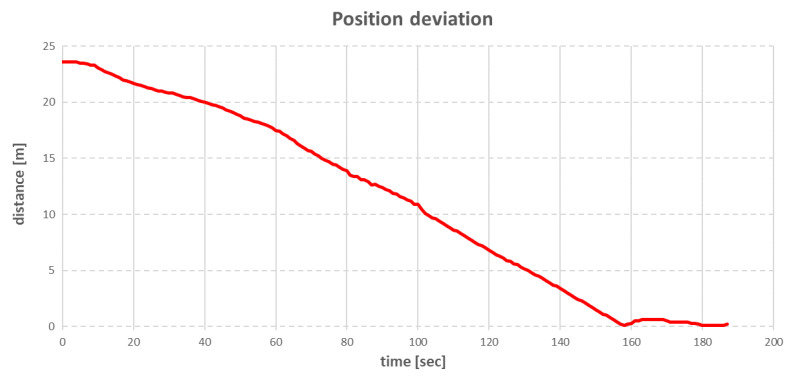
The ship’s position deviation after changing target position.

**Figure 15 sensors-22-07421-f015:**
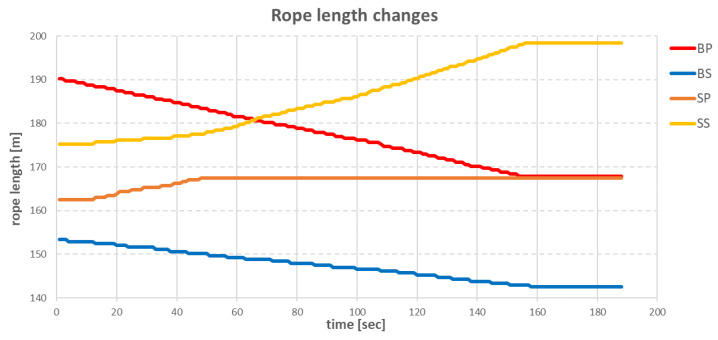
Changes in the length of individual ropes during the operation.

**Figure 16 sensors-22-07421-f016:**
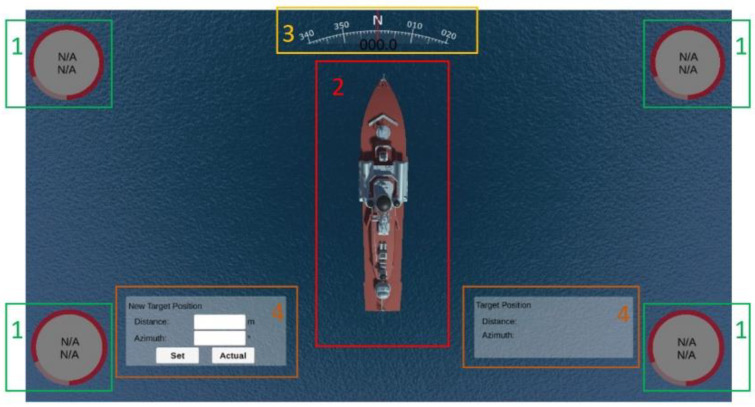
The user interface.

**Figure 17 sensors-22-07421-f017:**
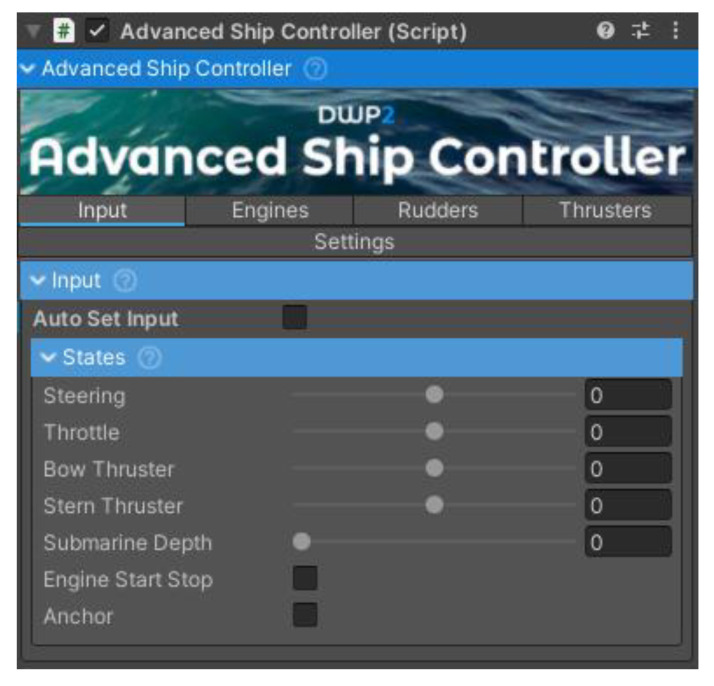
The “Advanced Ship Controller” component.

**Figure 18 sensors-22-07421-f018:**
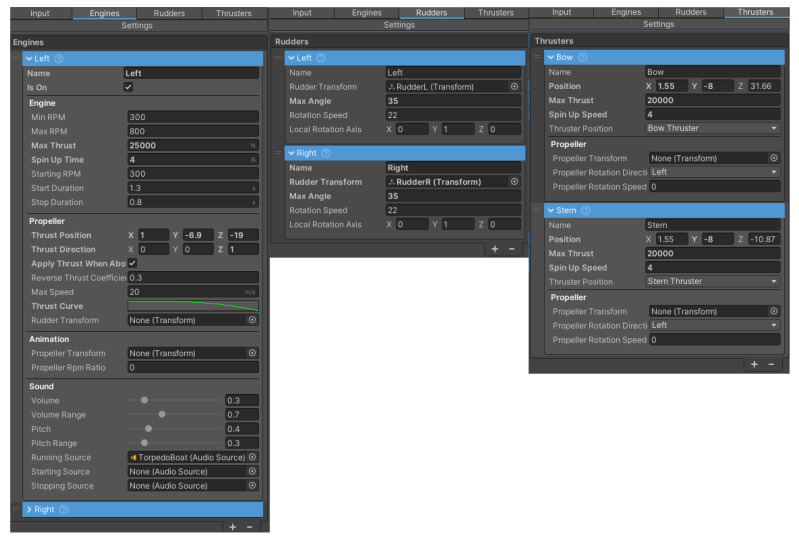
Tabs of the “Advanced Ship Controller” component.

**Figure 19 sensors-22-07421-f019:**
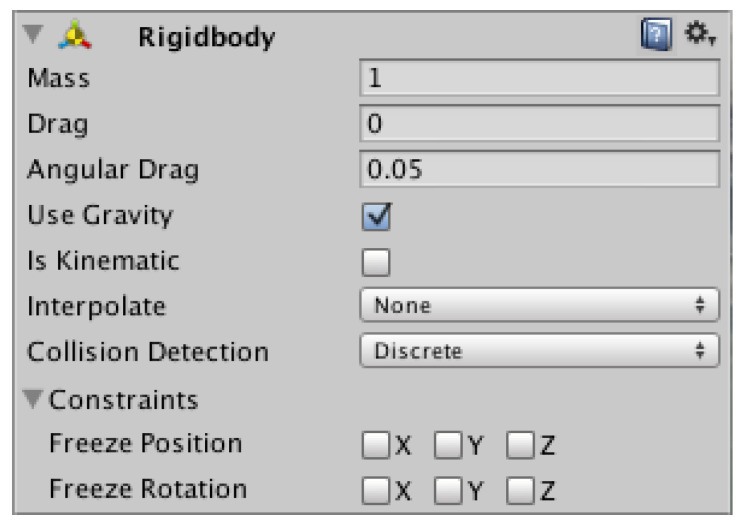
“Rigidbody” component.

**Figure 20 sensors-22-07421-f020:**
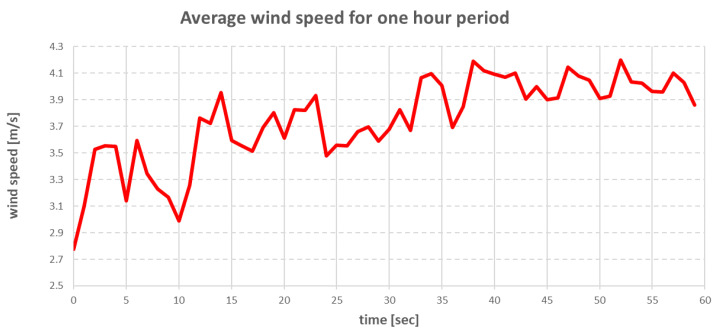
Wind speed changes during simulation.

**Figure 21 sensors-22-07421-f021:**
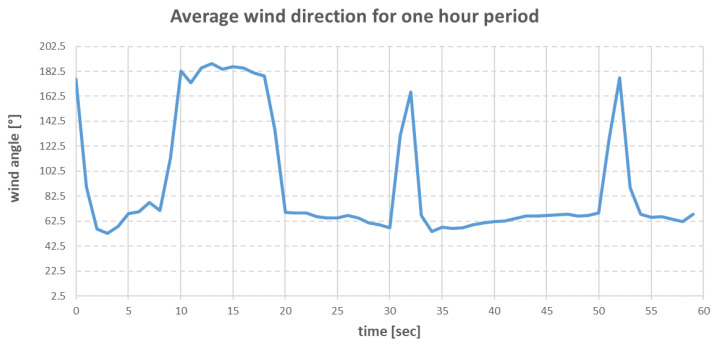
Wind direction changes during simulation.

**Figure 22 sensors-22-07421-f022:**
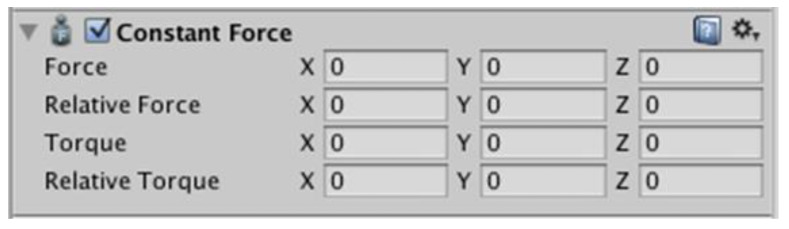
“Constant Force” component.

**Figure 23 sensors-22-07421-f023:**
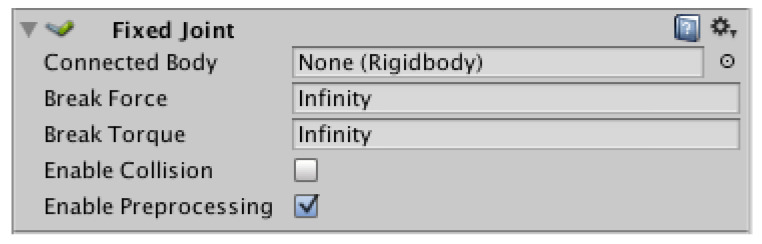
“Fixed Joint” component.

**Figure 24 sensors-22-07421-f024:**
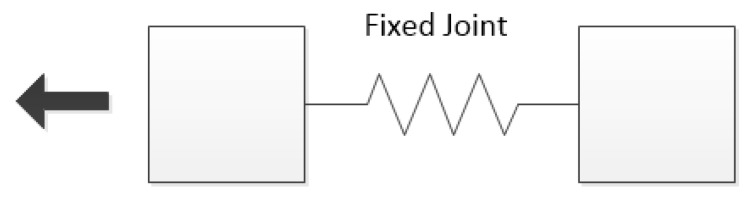
Illustrated operation of the “Fixed Joint” component.

**Figure 25 sensors-22-07421-f025:**
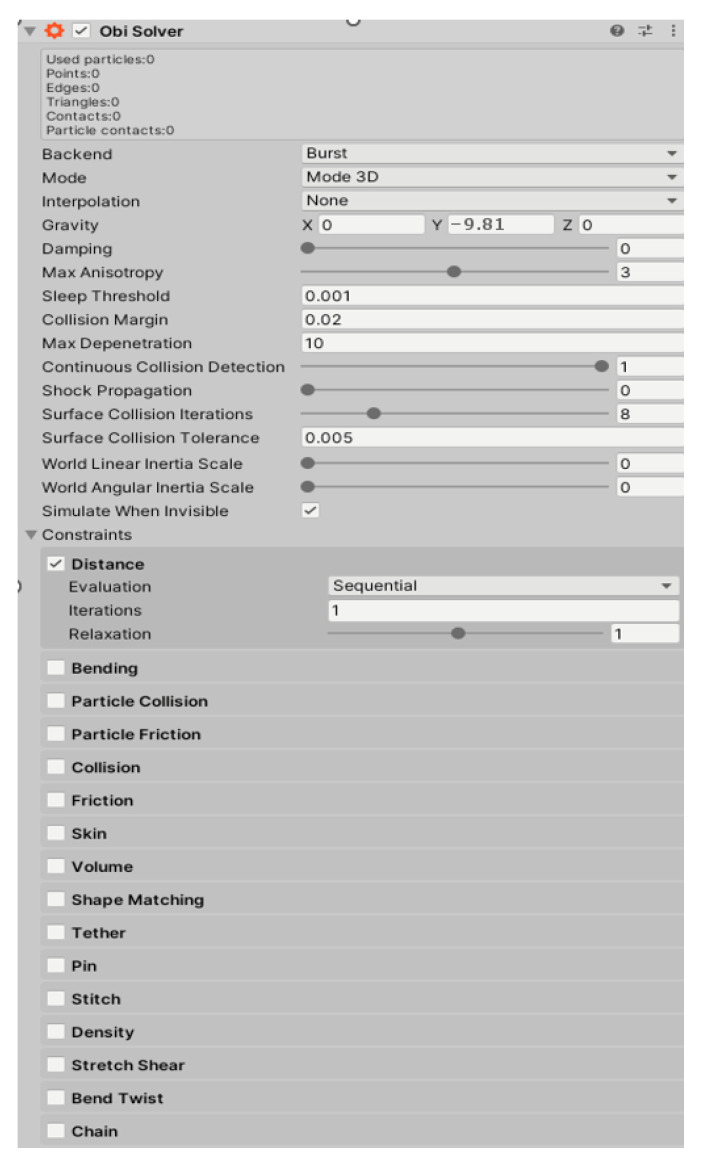
“Obi Rope” component.

**Figure 26 sensors-22-07421-f026:**
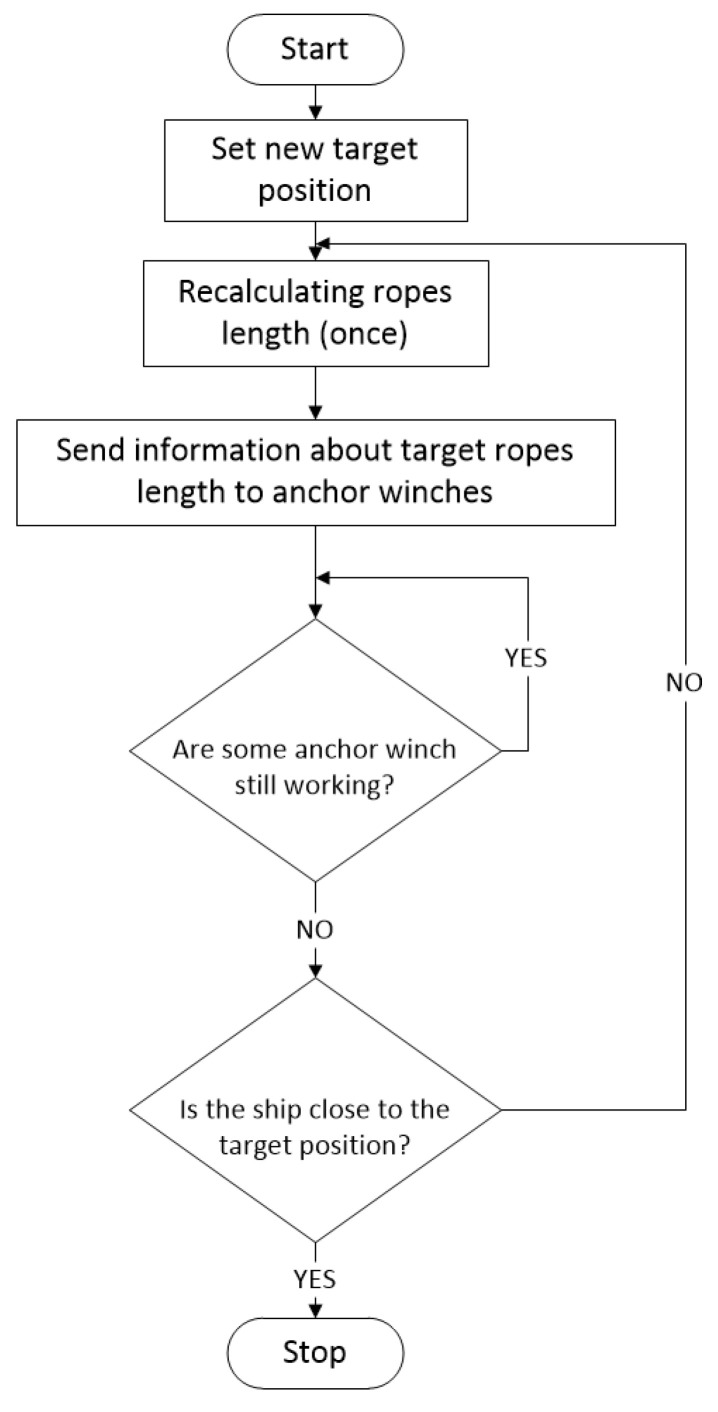
General control algorithm.

**Figure 27 sensors-22-07421-f027:**
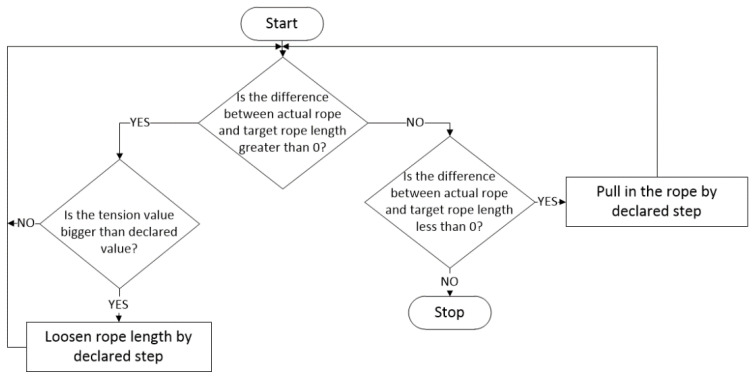
Anchor winch operation diagram.

**Figure 28 sensors-22-07421-f028:**
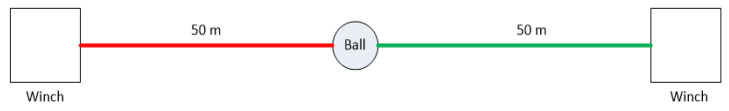
The first experiment.

**Figure 29 sensors-22-07421-f029:**
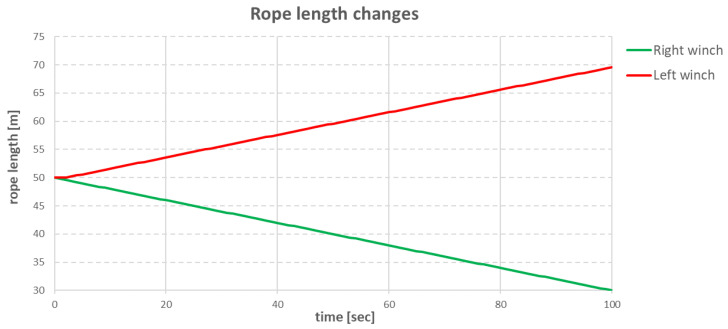
Changes in the length of individual ropes over time.

**Figure 30 sensors-22-07421-f030:**
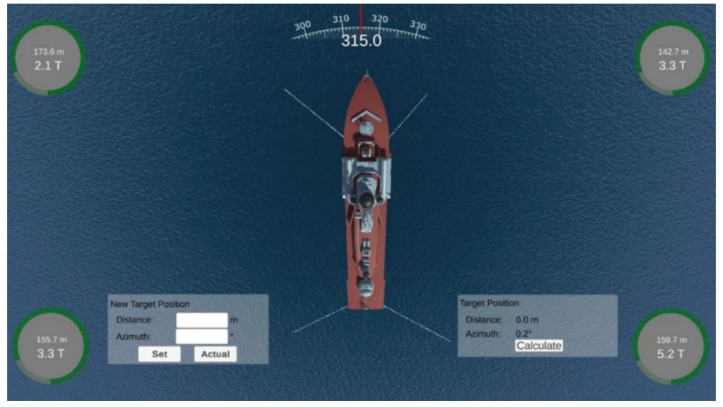
The anchorage in Unity3d.

**Figure 31 sensors-22-07421-f031:**
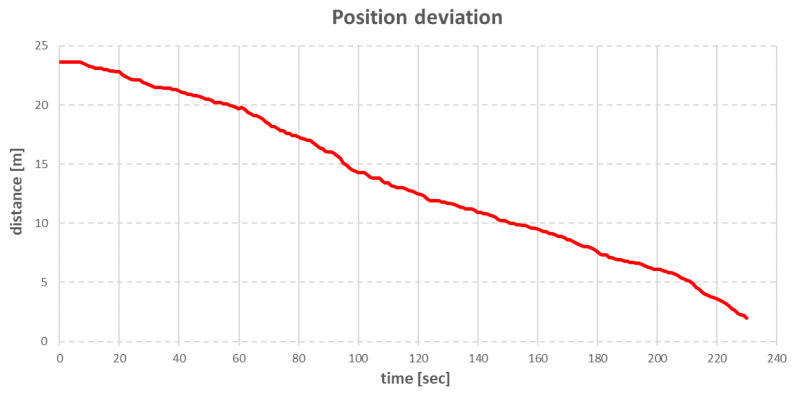
The ship’s position deviation after changing target position.

**Figure 32 sensors-22-07421-f032:**
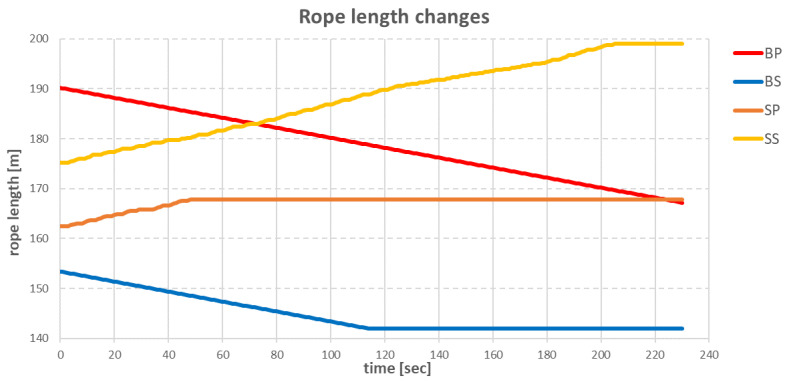
Changes in the length of individual ropes during the operation.

**Figure 33 sensors-22-07421-f033:**
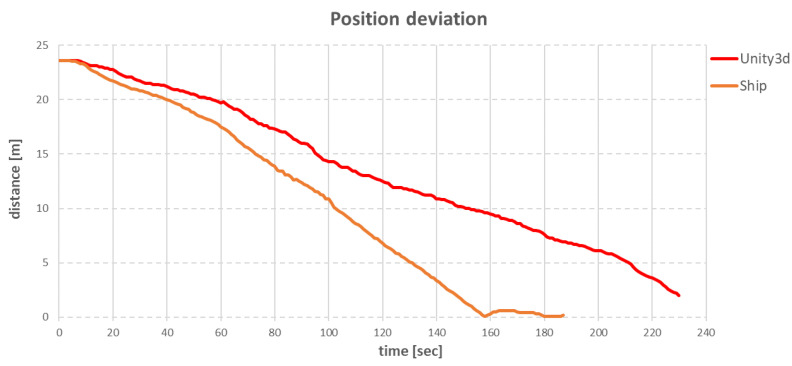
Comparison of the ship’s position deviation after changing target position, between the real ship and the virtual ship.

**Figure 34 sensors-22-07421-f034:**
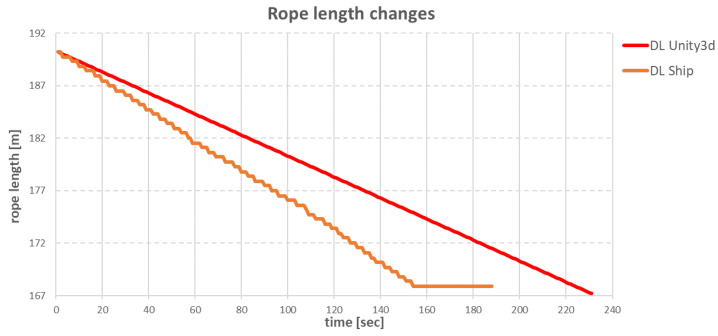
Changes in BP rope lengths over time.

**Figure 35 sensors-22-07421-f035:**
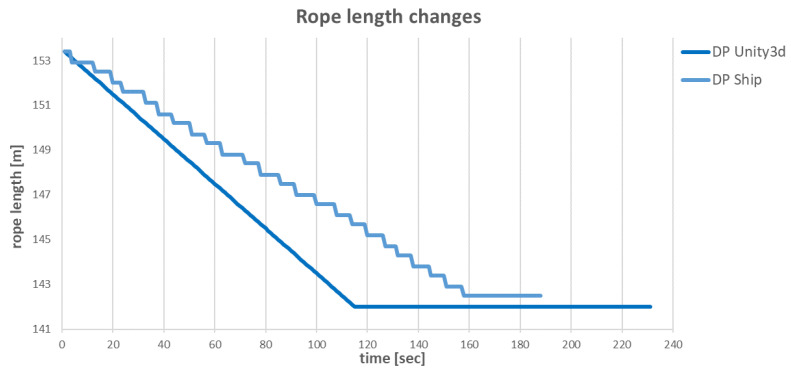
Changes in BS rope lengths over time.

**Figure 36 sensors-22-07421-f036:**
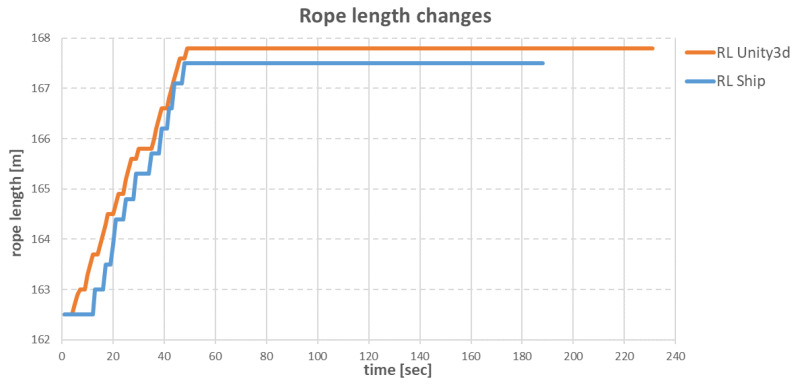
Changes in **SP** rope lengths over time.

**Figure 37 sensors-22-07421-f037:**
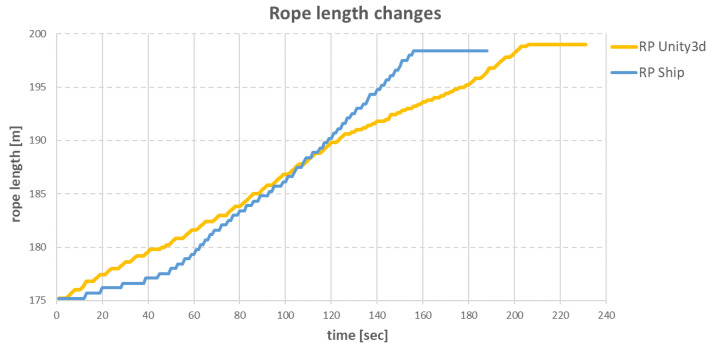
Changes in SS rope lengths over time.

**Table 1 sensors-22-07421-t001:** Units used in Unity3d.

Parameter	Unit
Mass	kg
Object speed	m/s
Wind speed	m/s
Environmental force	N
Thrusters force	N
Distance	M
Area	m^2^

**Table 2 sensors-22-07421-t002:** Ship dimensions.

Parameter	Value
Length overall (*L_oa_*) (m):	72.7
Length over spars (*L_os_*) (m):	68.3
Length between perpendiculars (*L_pp_*) (m):	64
Breadth (m):	11.6
Draught (m):	3.4
Displacement (T):	1886
Distance between fore-most and aft-most point of the hull below the surface at design draft even keel (m):	67.1
Water plane area (m^2^):	639
Projected longitudinal area above water (m^2^):	437
Surge position of geometric center of the projected longitudinal area above water with respect to *L_pp_*/2 (m):	0.1
Projected longitudinal area below water (m^2^):	223
Surge position of geometric center of the projected longitudinal area below water with respect to *L_pp_*/2 (m):	−2.9
Surge position of water line center with respect to *L_pp_*/2 (m):	−1.5
Projected transverse area above water (m^2^):	140
Projected transverse area below water (m^2^):	36

**Table 3 sensors-22-07421-t003:** Arrangement of anchor winches.

Anchor	X Position (m)	Y Position (m)
BP	32.1	2.7
BS	32.1	−2.7
SP	−34.8	2.9
SS	−34.8	−2.9

In Table 3, the anchor names mean: BP—anchor winch and port side bow anchor; BS—anchor winch and starboard side bow anchor; SP—anchor winch and port side stern anchor; SS—anchor winch and starboard side stern anchor.

**Table 4 sensors-22-07421-t004:** Distance from the vessel and the initial ropes lengths.

Anchor	Distance	Initial Rope Length
BP	173.6 m	189.7 m
DP	142.7 m	152.9 m
SP	155.7 m	162.5 m
SS	159.7 m	175.2 m

**Table 5 sensors-22-07421-t005:** Current length of individual anchor ropes and determined length of anchor ropes.

Anchor	Initial Rope Length	Calculated Rope Length	Δ Length
BP	189.7 m	167.2 m	−20.5 m
BP	152.9 m	142 m	−10.9 m
SP	162.5 m	168.3 m	+5.8 m
SS	175.2 m	199.4 m	+24.2 m

## Data Availability

Not applicable.

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
