# Peer review of "Verification of the System for Ship Position Keeping Equipped with a Set of Anchors in Unity3d"

_sensors, 2022, doi:10.3390/s22197421_

Round 1
Reviewer 1 Report
This article presents simulator implementation in the Unity3d game engine. The simulator emulates navigation devices like the GPS and Echosounder. The main goal of the simulation is to verify mathematical model of anchor winches. The model is developed based on research on real ship with anchor base positioning system. This paper can be reconsidered after revisions below.
1. In equation 17, what is the meaning of “L_{pp}-length between perpendiculars”?
2. “q” in equations 18 and 19 is different from “q” in the above, which is easy to be confused. The same problem is for “α” in equation 22.
3. In Figure 34, the maximum value of the abscess is around 230. How to see "while for the virtual ship, it took about 240 seconds"?
4. Some related literatures working on anchors-based positioning are recommended to be surveyed and compared, i.e., A Disaster Management-Oriented Path Planning for Mobile Anchor Node-Based Localization in Wireless Sensor Networks; Connectivity-Based Localization Scheme for Social Internet of Things.
5. Why does the positioning system choose four anchors instead of three?
6. How to reflect the superiority of the proposed algorithm by experiments?
Author Response
Reviewer No 1
This article presents simulator implementation in the Unity3d game engine. The simulator emulates navigation devices like the GPS and Echosounder. The main goal of the simulation is to verify mathematical model of anchor winches. The model is developed based on research on real ship with anchor base positioning system. This paper can be reconsidered after revisions below.
Dear Reviewer 1,
I would like to express my gratitude for the time devoted to the review of our article. The comments made by the respected Reviewer will certainly improve the quality of the content presented and the legibility of the proposed publication. The main corrections in the paper and our responds to the Reviewer’s comments are as following:
- In equation 17, what is the meaning of “L_{pp}-length between perpendiculars” ?
Ad.1.
We would like to thank the respected reviewer for pointing out that not all abbreviations that we used are understandable for everyone. Lpp is the distance between the ship's aft perpendicular and the fore perpendicular. Approximately this is the length of the submerged part of the ship's hull. In order to better illustrate what the Lpp length means, we have added Figure 2.
- “q” in equations 18 and 19 is different from “q” in the above, which is easy to be confused. The same problem is for “α” in equation 22.
Ad.2.
We are very grateful to the respected reviewer for this comment. Indeed, some notations could be misleading, so we decided to introduce uniform notations in equations 15-36, e.g. for "q" in the case of wind force, it will now be "qw", and for "q" in the case of sea current, "qc". The rest of the equations were corrected in the same way.
- In Figure 34, the maximum value of the abscess is around 230. How to see "while for the virtual ship, it took about 240 seconds"?
Ad.3.
We are very appreciative of the respected reviewer for this remark. The value 240 appears due to rounding to the full decimal value. The actual value is 233. In accordance with the remark of the respected reviewer, we changed the value given in the article to the real one and for better readability we corrected the scale in all figures.
- Some related literatures working on anchors-based positioning are recommended to be surveyed and compared, i.e., A Disaster Management-Oriented Path Planning for Mobile Anchor Node-Based Localization in Wireless Sensor Networks; Connectivity-Based Localization Scheme for Social Internet of Things.
Ad.4.
We are thankful to the respected reviewer for this remark. Currently, the entire chapter 1 (introduction) has been rewritten. We removed redundant drawings and changed the whole content. As suggested by the respected reviewer, we focused on comparing existing reference systems, such as radio, radar and acoustic systems, but also mentioned less popular systems, e.g. laser positioning system. In the descriptive part of the algorithm, we compared the current reference systems.
- Why does the positioning system choose four anchors instead of three?
Ad.5.
We thank the respected reviewer for this question. Due to the use of 4 anchors, the ship is able to keep its position regardless of the direction of the impact of wind, sea current and waves. Additionally, with 4 anchors, there is no limitation in the direction of change of the target position. When positioning with 3 anchors, the entire positioning operation must be carefully elaborated and thought out, as some position changes may be impossible to do.
- How to reflect the superiority of the proposed algorithm by experiments?
Ad.6.
We kindly appreciate the respected reviewer for this question. The goal of the publication was to present the developed simulator for the purposes of research on the anchor-base positioning system. According to the actual state of knowledge, there is a lack of materials describing this type of systems in the literature, which focus directly on changing the position of the ship with the use of a set of anchors. The development of mathematical models and the simulation environment will allow further research on algorithms related to the optimization of the anchor placement and ship positioning process. This issue concerns both the order in which anchors are placed and the movement of the ship on already placed anchors. In addition, it is possible to analyse the various hydrometric conditions in the presence of which positioning takes place. This will make it possible to check the manoeuvrability of a ship using the anchor system and to determine the limit values for hydrometeorological conditions in which the system can be used.
Thank You very much once again the Respected Reviewer for his comments and time.
Best Regards
Jakub Wnorowski
Andrzej Łebkowski

Reviewer 2 Report
# Suggestions for improvement
However, they do not summarize their research's major findings and contributions! The section needs to be rewritten.
The related works section needs to be reviewed again.
Reconstruct the abstract and add motivation and novelty to the proposed method. I suggest comparing some similar works.
Several evaluation criteria should be included for performance comparison.
Evaluate overhead using the evaluation criteria. Choose an optimal method of evaluating overhead. The implementation details should be clearly specified, if possible.
What is the throughput of the system? How about the delay?
There is no computational review.
It is not considered whether implementation is completed properly.
Add details about time and space complexity
Your audience will understand your content better if you include a flowchart in your research.
Author Response
Reviewer No 2
# Suggestions for improvement
Dear Reviewer 2,
Thank You very much for posting all the remarks and comments. Based on your comments, we have made extensive modifications to the original manuscript. Those comments are all valuable and very helpful for revising and improving our paper, as well as the important and significance guide to our research. I hope that the implemented changes (based on comments from all Reviewers) will satisfy the Reputable Reviewer enough to accept the proposed paper for publication.
- However, they do not summarize their research's major findings and contributions! The section needs to be rewritten.
Ad.1.
We thank the respected reviewer for the comment made. Currently, Chapter 5 (summary) has been completely restructured. We hope that the content presented summarizes the research carried out representing the main discoveries and contributions of the research to the development of science.
- The related works section needs to be reviewed again.
Ad.2.
We are grateful to the respected reviewer for his attention. As suggested, the entire Chapter 1 and abstract have been thoroughly revised. Redundant figures were removed and the content was modified. This time we focused on comparing existing reference systems, such as radar or acoustic systems, but also mentioned less popular systems, e.g. laser positioning system. In the descriptive part of the algorithm, we compared actual reference systems.
- Reconstruct the abstract and add motivation and novelty to the proposed method. I suggest comparing some similar works
Ad. 3.
We would like to thank the respected reviewer for this suggestion. The abstract has been modified accordingly as suggested.
- Several evaluation criteria should be included for performance comparison.
Ad. 4.
We want to thank the respected reviewer for his attention. In order to determine the performance of a positioning system, you can use multiple evaluation criteria, depending on which aspect you are interested in, such as.
- A criterion related to energy consumption during operations. In this case, compare the energy consumption of a dynamic positioning system and positioning using a set of anchors. In the first case, the energy consumption will be very high, because the thrusters operate during the whole period of the operation, whether the ship is moving or not. In the second case, if the ship is not moving, it does not consume energy for positioning purposes, because only the anchors keep the ship in the set position.
- A criterion related to computer system performance. In this case, it should be taken into account if calculations related to rope length calculations, communication with ship sensors and communication with controllers affect positioning delays, if the computer at the operator's station freezes, etc. We described more on this subject when answering a question from the respected reviewer about system performance.
- A criterion related to positioning accuracy. In this case, for a vessel using anchor-base positioning, the evaluation is more difficult because there are no defined rules and assumptions to refer to. If one considers the requirements that are placed on ships equipped with a DP system (which by definition should be more accurate than a system based on a set of anchors), when inspecting DP systems, one of the evaluation criteria is the deviation from the target position after changing ship position. It is assumed that a DP system will pass the test if the deviation is less than 5 meters. In our case, the deviation from the target position was between 0.6 and 0.1 meters. The evaluation criterion for position accuracy was described by us in Chapter 5 (summary).
- Evaluate overhead using the evaluation criteria. Choose an optimal method of evaluating overhead. The implementation details should be clearly specified, if possible.
Ad. 5.
We appreciate the respected reviewer for this comment. Unfortunately, we declared with the owner of the vessel on which the research was carried out (a military facility) that some sensitive data will not be presented in the article. Such data are, for example: the name of the ship, photos of the ship, the precise geographic position, the location of the research, but also details related to the implementation of the system. We could have included a general description of the system in the article, but without the details. We hope that the respected reviewer understands the reasons for this decision.
- What is the throughput of the system? How about the delay?
Ad. 6.
We would like to thank the respected reviewer for this question. The reading of information from the navigation equipment was once per second. Refreshing of data in the user interface was every 500 ms, while the ship's course data was transmitted 130 times per second. Information exchange with individual PLCs was done every 200 ms. Thanks to the use of separate threads for each navigation device and PLC, such frequent information readings did not affect the system's performance. The answer to this question is also included in the text of the article, in Chapter 3.
- There is no computational review.
Ad. 7.
We appreciate the respected reviewer for bringing this aspect to our attention. In the corrected version of the article we have added information on the reasons for the differences between the results obtained during the research on the real ship and during the verification of the mathematical model of the anchor winches. The list of reasons by which the differences occurred is as follows:
- Too perfect representation of the work of anchor winches in the mathematical mode.
- The anchor ropes in the mathematical model had a stretching effect because non-stretchable ropes could not be obtained in Unity3d.
- In the simulator, there is greater precision in the calculation of geographic coordinates. On the real ship, the system operated on the geographic position obtained from the GPS with an accuracy of up to 1 [m], while the simulator determines the geographic position with a resolution of 1 [mm]
The performance of the application (user interface) located at the real ship's operator station can become an additional element of evaluation. The performance of the system is most simply represented as the value of consumed computer resources. In our case, throughout the application's operation, the user interface used 400 MB of RAM and loaded the CPU at 30%. We noticed an increase in RAM consumption only after creating new threads for each navigation device and PLC. Before the threads were created, the used RAM was 200 MB.
- It is not considered whether implementation is completed properly.
Ad. 8.
We would like to thank the respected reviewer for paying attention. In the end, we were able to implement the algorithm we described on the ship where the research was carried out. We have added information about the implementation of the system in the article's summary.
- Add details about time and space complexity.
Ad. 9.
We want to thank the respected reviewer for this suggestion. In Chapter 3, we have added details of the time the ship reached the target point and information on the changes in the distance between the ship and the target point. The description is under figure 14.
- Your audience will understand your content better if you include a flowchart in your research.
Ad. 10.
We are grateful to the respected reviewer for this suggestion. Of course, we agree that some of the elements included in the article might not be understandable on the basis of the presented textual content, therefore we additionally prepared a diagram of the information flow in the system (Fig. 12). The figure shows what happens in the system after the new ship position is set.
Dear Reviewer, once again, we would like to thank you very much for your comments, which contributed significantly to the improvement of the scientific level of our manuscript.
Best Regards
Jakub Wnorowski
Andrzej Łebkowski

Reviewer 3 Report
Various parts of the manuscript needs to be reconsidered:
- The use of English language needs to be reconsidered: grammar, punctuation, spelling and overall style.
- Abstract of the manuscript is too limited. The introductional part (beginning) of the abstract needs to be extended.
- Few more keywords could be added. For example, keyword "Ship" could be added.
- Structure of the manuscript must be reconsidered. Introduction of the manuscript (section 1) is very long, there are a lot of excessive figures.
- Major drawback of the manuscript – literature review. In general, there is no proper literature review, which would analyse scientific research works.
- How proposed mathematical model is more advanced, while in comparison with other similar models?
- Conclusions of the manuscript do not properly represents the performed research.
Author Response
Reviewer No 3
Various parts of the manuscript needs to be reconsidered
Dear Reviewer 3,
Thank You for taking the time to review our manuscript titled " Verification of the system for ship position keeping equipped with a set of anchors in Unity3d". The comments provided are valuable and very helpful in improving our article, as well as an important guiding for our research.
Responses to the detailed comments are provided below:
- The use of English language needs to be reconsidered: grammar, punctuation, spelling and overall style.
Ad.1.
We would like to thank the respected reviewer for this remark. Indeed, there were various types of typos in the text, which we tried to catch and correct. We checked the text several times in terms of grammar, punctuation, spelling and overall style. We hope that the current version will meet the expectations of the respected reviewer.
- Abstract of the manuscript is too limited. The introductional part (beginning) of the abstract needs to be extended.
Ad.2.
We are very grateful to the respected reviewer for this remark. In accordance with the remark, we have expanded the beginning of the abstract by adding our motivation for carrying out the research. We have also expanded the abstract, with a more detailed description of the system proposed in the article.
- Few more keywords could be added. For example, keyword "Ship" could be added
Ad. 3.
We are very appreciative of the respected reviewer for this advice. We have added the following keywords: marine systems and ship anchor winch modelling
- Structure of the manuscript must be reconsidered. Introduction of the manuscript (section 1) is very long, there are a lot of excessive figures. Major drawback of the manuscript – literature review. In general, there is no proper literature review, which would analyse scientific research works.
Ad. 4.
We kindly appreciate the respected reviewer for this comments. Now, the whole Chapter 1 (Introduction) has been completely rebuilt. We have removed unnecessary figures and changed the entire content. As suggested, we have focused on the comparison of existing reference systems. In the descriptive part of the algorithm, we compared the current positioning systems with the system using anchors. Answers to all comments have been inserted into the text of the proposed manuscript.
- How proposed mathematical model is more advanced, while in comparison with other similar models?
Ad. 5.
We are thankful to the respected reviewer for this question. According to the current state of knowledge, there are no publications in the literature treating anchor-base positioning systems for changing the position of a ship. It is possible to find publications that mention, for example, the optimal placement of anchors to negate the effect of wind force on the vessel, but these usually apply to static vessel such as an oil rig. With this in mind, it is difficult for us to answer the respected reviewer's question about the advancement of our model compared to others.
Thanks to the development of the mathematical model, it is possible to perform all work related to the planning and placement of anchors, as well as the possibility of moving the ship using a set of anchors.
- Conclusions of the manuscript do not properly represents the performed research.
Ad. 6.
We would like to thank the respected reviewer for the presented remark. Chapter 5 (Summary) has been thoroughly revised as suggested. We hope that the presented content sufficiently presents the research carried out and the analysis of its results.
Thank You very much again for Your comments and Your time.
Best Regards
Jakub Wnorowski
Andrzej Łebkowski

Round 2
Reviewer 1 Report
No further question.
Author Response
Reviewer No 1
No further question.
Dear Reviewer 1,
Thank You very much once again the Respected Reviewer for his comments and time.
Best Regards
Jakub Wnorowski
Andrzej Łebkowski

Reviewer 2 Report
The authors revised the paper carefully.
Author Response
Reviewer No 2
The authors revised the paper carefully.
Dear Reviewer 2,
Thank You very much once again the Respected Reviewer for his comments and time.
Best Regards
Jakub Wnorowski
Andrzej Łebkowski

Reviewer 3 Report
Manuscript sill could be improved:
- In my opinion, literature review is still a major drawback of the manuscript. As noted previously, there is no proper detailed literature review, which would analyse in detail scientific research works.
- Why exactly ship dimensions, defined in table 2, was selected and used? The assumptions / conditions of the verification process should be explained more in detail / more clearly.
- What are the possible drawbacks of the proposed methodology?
Author Response
Reviewer No 3
- In my opinion, literature review is still a major drawback of the manuscript. As noted previously, there is no proper detailed literature review, which would analyse in detail scientific research works.
Ad.1.
We would like to thank the respected reviewer for this comment. We are sorry that the literature review has not been properly revised. In the current version of the article, Chapter 1 has been revised to describe what is included in each reference. In addition, we have added new references related to anchors and ships using anchors for positioning. We sincerely hope that this time the literature review will be accepted by the respected reviewer.
- Why exactly ship dimensions, defined in table 2, was selected and used? The assumptions / conditions of the verification process should be explained more in detail / more clearly.
Ad.2.
We appreciate the respected reviewer for this question. Indeed, the wording we used in the article may have been misleading. Table 2 shows the dimensions of the real ship on which the research was carried out. The data from the table was used to create a 3d model of the ship and to model it in Unity3d. The ship with the dimensions given in Table 2 was chosen for the research because the ship's owner would like to implement an automatic anchor positioning system on it and his other ships (of the same type) in the future.
The assumptions of verification depend on what part of the research process interests us:
- Verification assumptions related to the control algorithm:
- The deviation of the ship's position after changing the target point should be less than 5 metres (DNV GL recommendations for DP systems).
- The research was to be carried out on 4 anchors.
- Verification assumptions related to the mathematical model of anchor winches:
- Mapping in Unity3d the environmental conditions that took place during the research on the real ship.
- Mapping in Unity3d the dimensions of the real ship on which the research was carried out.
- Mapping the geographic coordinates of the real ship and the geographic coordinates of the dropped anchors.
- Mapping in the mathematical model of anchor winches the principle of their functioning based on research carried out on the real ship and implementing it in Unity3d.
- Mapping in Unity3d the ship position shift that was performed on a real ship.
- What are the possible drawbacks of the proposed methodology?
Ad. 3.
We are grateful to the respected reviewer for this question.
Disadvantages associated with the anchors-based positioning method presented in the publication include:
- It is not possible to use the positioning algorithm when the ship has only placed 2 anchors. In such a situation, stabilisation of the ship's position is ineffective.
- Changing the ship's course is possible, but only if 4 anchors are placed.
- The process of changing position is slow. The speed of movement of the real ship was approximately 0.1 m/s.
As general disadvantages of anchor-based positioning, we can include:
- The long process of placing anchors.
- Cannot be used in close range to other objects.
- Throughout the positioning period, observe that other vessels do not damage the anchor ropes that have been deployed.
In addition to the disadvantages outlined above, the following advantages of the system can also be highlighted:
Advantages associated with the anchors-based positioning method presented in the publication include:
- Reducing the number of crew needed to operate.
- Improving safety on board.
- Automation of the anchors’ deployment process.
- Reduced time required to deploy anchors.
- Automation of the process of changing the ship's position.
- No need to use reference systems to keep position.
As general advantages of anchor-based positioning, we can include:
- Zero / low energy consumption during positioning / changing position compared to DP systems.
- A simpler mathematical model compared to DP systems.
- Better position stability in case of frequent changes in environmental conditions compared to DP systems.
- High level of safety for those working under and around the ship (e.g. divers) due to the lack of moving parts
We would like to thank the respected reviewer for all comments. We deservedly receive each of them. We are also grateful for the fact that the honorable reviewer showed us the elements that we could introduce to raise the scientific value of the article. The fact that we were given such detailed comments proves the full professionalism of the reviewer. We hope that our answers and the changes introduced in the publication will be satisfactory.
Best Regards
Jakub Wnorowski
Andrzej Łebkowski
